# Minimizing Cost Overrun in Rail Projects through 5D-BIM: A Systematic Literature Review

Osama A. I. Hussain [1], Robert C. Moehler [1,*], Stuart D. C. Walsh [1] and Dominic D. Ahiaga-Dagbui [2]

1   Department of Civil Engineering, Faculty of Engineering, Monash University, Melbourne, VIC 3800, Australia; osama.hussain@monash.edu (O.A.I.H.); stuart.walsh@monash.edu (S.D.C.W.)
2   School of Architecture and Built Environment, Deakin University, Geelong, VIC 3220, Australia; dominic.ahiagadagbui@deakin.edu.au
*   Correspondence: robert.moehler@monash.edu

**Abstract:** Mega projects delivering rail infrastructure are constantly seeking cost-effective and efficient technologies to sustain the growing population. Building information modeling (BIM) and BIM for cost management (5D-BIM) have shown great potential in the building industry and have been adopted widely. However, 5D-BIM implementation in rail infrastructure is still in its infancy. This paper presents a systematic literature review of 380 publications related to cost overrun, cost management and 5D-BIM for rail infrastructure, including rail projects. The review identified knowledge gaps and synthesized existing research on cost overrun in rail projects, cost estimation models, and the current use of 5D-BIM. The review revealed that there is no current study integrating 5D-BIM into the rail project lifecycle. This paper highlights the importance of integrating 5D-BIM systematically in the rail project life cycle to avoid/minimize cost overrun. The review provides researchers and practitioners with crucial information for deploying 5D-BIM to minimize cost overruns in rail projects.

**Keywords:** 5D-BIM; cost management; cost overrun; rail projects; systematic literature review (SLR); mega-projects

## 1. Introduction

The promotion of sustainable long-term economic and social development in societies is greatly influenced by infrastructure mega-projects. These projects have the potential to shape the national economy and boost GDP. However, the benefits of these projects come with a high risk of failure. Even though projects may perform well technically, poor cost performance may jeopardize the project's existence or its economic justification [1,2]. The detrimental consequences of cost overrun are widely acknowledged among academics, despite ongoing debates on its definition, causes, magnitude, and reference points for measurement [3–10]. A study by Flyvbjerg [11] found that nine out of ten infrastructure mega-projects go over budget. Rail projects, for example, go over budget by an average of 44.7 percent. The cost of these projects justifies the relentless pursuit to avoid cost overrun, as it is not unusual for rail projects to cost USD 100 billion [12] or more [13].

Different parties use various frameworks, tools, and strategies for project governance to address the cost overrun phenomenon. Government agencies/clients focus on preventing cost overrun on the project life cycle level, while contractors, consultants, and operators are concerned about specific project stages. Despite the calls for digitalization in construction and the use of emerging technologies such as building information modeling (BIM), the application of these technologies on the project life cycle level is still limited, and project governance has yet to benefit from technological advancements [14,15].

Effective project governance is essential for the successful delivery of complex and complicated rail projects [16]. These projects demand a wide range of expertise across multiple fields, such as engineering, construction, urban planning, and transportation policy. Moreover, the financing models for these projects are often intricate, involving a mix

of public and private funding sources, including tax revenues, government grants, loans, and private sector investments, which further adds to the complexity. Additionally, managing a diverse array of stakeholders, including government agencies, local communities, businesses, and interest groups, is a crucial aspect of these projects [17,18].

The 3D-BIM model used in rail projects can be further enriched by the addition of a wide range of graphical and non-graphical information, such as geographic information system (GIS) data, asset tagging, and maintenance requirements. The 4D-BIM model incorporates time-related information, providing a more complete understanding of the project schedule, while the 5D-BIM model encompasses cost-related information.

5D-BIM offers a comprehensive and collaborative approach for cost management and control, as well as financial decision-making support throughout the project lifecycle [19,20]. Successful implementation of 5D-BIM requires a thorough understanding of the causes of cost overrun phenomena, current cost management and control strategies in transportation/rail and their limitations, the models used for cost estimation, and an understanding of the current uses of 5D-BIM.

The existing literature indicates a noteworthy research gap regarding the integration of 5D-BIM in rail projects. Previous studies have primarily focused on the application of 5D-BIM in other industries, such as construction, with inadequate consideration given to its implementation in rail projects. In light of the potential benefits of integrating 5D-BIM into rail projects' lifecycles [21–23], this paper presents a novel approach to exploring the current application of 5D-BIM in rail projects by using the systematic literature review (SLR) research methodology to determine the current state of 5D-BIM application in rail projects, identify research gaps, and directions for future research.

The SLR is a rigorous and comprehensive method of collecting, evaluating, and synthesizing existing literature on a particular research question or topic. This process involves identifying, selecting, and evaluating the quality of relevant studies and synthesizing the findings to identify trends and gaps in the literature. The SLR follows a structured approach to minimize bias and ensure result replication [24]. The study addresses the following four research questions:

1. What causes cost overrun in transport projects in general and rail projects in particular?
2. What are the cost models used to predict and analyse cost overrun in transport projects in general and rail projects in particular?
3. What cost management and control strategies are used to prevent these cost overruns? What is the efficiency of these strategies and suitability for 5D-BIM modelling?
4. How can 5D-BIM be successfully integrated into rail projects life cycle to support cost management and control models and minimize/prevent cost overrun?

## 2. Background and Terminology

This section outlines the key terminology and concepts, including infrastructure mega-projects and its characteristics, the rail industry and its common terminology, cost overrun definitions, BIM dimensions, cost management and its functions: cost estimation, modelling, and cost budgeting, as well as the various techniques used for cost/budget monitoring and control.

### 2.1. Infrastructure Mega-Projects

The majority of rail projects are indeed mega-projects (or major projects); therefore, it is necessary to understand that they share the same characteristics/challenges/problems by definition.

The definition of mega-projects in the literature is inconsistent, with some sources using the term interchangeably with "large projects" or major projects [25]. Ruuska et al. [26] define mega-projects as complex undertakings involving multiple organizations with different objectives that can have significant socio-political implications. Capka [27] describes them as expensive projects requiring the management of numerous and complicated activities while adhering to strict deadlines and budgets. Flyvbjerg [11] differentiates "major

infrastructure projects" from "mega-projects" based on their estimated dollar value, limiting the former to hundreds of millions of dollars and the latter to more than USD one billion. Mega projects pose unique technological, sustainability and acceleration of delivery challenges due to their vast array of stakeholders and associated communication dynamics [28].

Chang et al. [29] distinguish Infrastructure mega-projects by their complexity, ambiguity, and the need for the integration of a large number of units over a long period of time. Mega-projects are distinct from other projects in five key elements, including a budget exceeding USD 500 million, complexity, uncertainty, dynamic interfaces, and running for a period that exceeds the technology cycle time of the technologies involved, attracting high levels of public and political interest, and being defined by effect rather than solution.

### 2.2. Infrastructure and Rail Projects

Infrastructure projects refer to the tangible assets built for public benefit, including public transportation systems such as rail transit, airports, highways, hospitals, energy and power, and water and wastewater facilities [30,31]. The rail transit system serves as a crucial component of modern cities' public transportation networks [32]. Rail transportation, a form of terrestrial-guided mass transport, can be categorized based on traction power, traffic volume, track type, and speed, as shown in Figure 1.

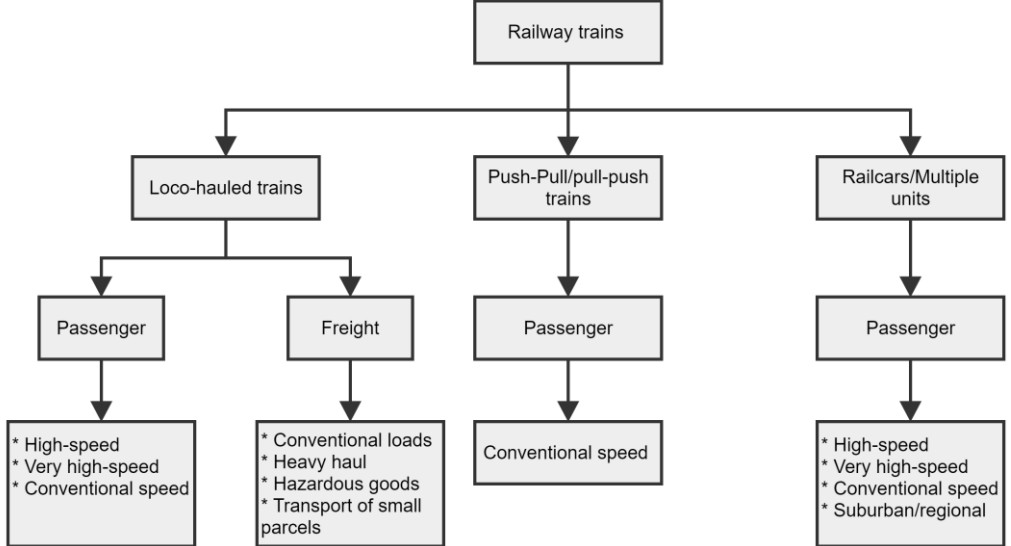

**Figure 1.** Types of Trains [33] Copyright 2022 by Taylor and Francis Group. Reprinted with permission.

The term "railway infrastructure" refers to the railway track, civil engineering structures, systems, and premises necessary for railway traffic [33]. In the US, this is referred to as "railroads" [34]. The rail network consists of tracks (links) and stations (nodes) for traffic transportation [35].

Compared to other construction projects, rail projects are known to be risky [36], both complex and complicated [16], and require efficient stakeholder communication and management [37]. While complicated projects are large and have highly predictable processes [16], complex projects are characterized by unpredictable and ever-changing processes and a delicate political, social, and economical stakeholder environment that can challenge project decisions and strategies [16,38,39]. The complexity of rail projects is mainly attributed to dimensions such as project finance, context, and site, which are outside of project control, and project management, delivery, and tasks, which are internal factors [16]. Additionally, the long construction cycle (up to 50 years) and the sophisticated electromechanical and signaling systems, as well as the high costs (hundreds of billions of dollars) involved, add to the complications associated with rail projects [18].

### 2.3. Cost Overruns

Cost overruns in infrastructure projects, particularly rail projects, result from uncertainty and misinformation surrounding project costs, benefits, and risks, leading to poor decision-making. This phenomenon is referred to as cost escalation, cost development, or cost increase by different authors [40,41]. Cost overrun refers to the difference between actual and estimated project costs [42]. Flyvbjerg [40] defines cost development as the difference between actual and projected costs as a percentage of projected costs, while cost escalation is defined as actual costs minus estimated costs as a percentage of estimated costs.

However, the definition of cost overrun remains a controversial issue in academic circles, as the reference point to measure cost overrun is a point of disagreement. Some authors, such as Flyvbjerg [40], use the budget estimate at the time of the decision-to-build as the reference point, while others, such as Bolan [43] and Ahiaga-Dagbui and Love [9], argue that this is an interim estimate that should not be used to evaluate cost performance. Infrastructure mega-projects have long life spans and obtaining planning permits can take up to 10 years [44]. During this time, changes in project scope, market conditions, and delivery methods can significantly alter cost estimates, which should not be considered as cost overruns [45].

The stakeholder perspective plays a crucial role in determining what is considered cost overrun, and this should be taken into consideration when reviewing and analysing the literature. For example, politicians may understand that preliminary estimates are unreliable, but still see them as acceptable risk compared to the benefits and overall impact of the project (see Ahiaga-Dagbui and Love [9]).

### 2.4. Five-Dimensional Building Information Modelling (5D-BIM)

Building information modeling (BIM) is a digital representation of an engineering project's entity and functional characteristics [46]. This approach encompasses the entire building project lifecycle, including planning, design, construction, operation, maintenance, and demolition, and facilitates as a project artefact the collaboration, the storage, sharing, exchange, and management of multidisciplinary information among stakeholders [47–49].

BIM is more than just software, it is a set of data sources and tools that support various disciplines and build a multidimensional virtual environment for the built environment [50]. The nD in BIM represents the number of dimensions linked to the virtual building model [51,52]. Table 1 shows the different characteristics of BIM dimensions.

**Table 1.** Characteristics of BIM dimensions. Adopted from [52] under Creative Commons licence (CC-BY 3.0).

| BIM Dimension | Descriptions | Characteristics |
|---|---|---|
| 3D | Geometry dimensions | 3D building data and information, field layout and civil data, reinforcement and structure analysis, existing model data. |
| 4D | 3D + Scheduling data (time) | Project schedule and phasing, just-in-time schedule, installation schedule, payment visual approval, last planner schedule, critical point. |
| 5D | 4D + Cost data | Conceptual cost planning, quantity extraction to cost estimation, trade verification, value engineering, prefabrication. |

**Table 1.** *Cont.*

| BIM Dimension | Descriptions | Characteristics |
| --- | --- | --- |
| 6D | 5D + Sustainability data | Energy analysis, green building element, green building certification tracking, green building point tracking. |
| 7D | 6D + Lifecycle info (operation and maintenance) | Building life cycles, BIM as built data, BIM cost operation and maintenance, BIM digital lend lease planning. |

The 4D-BIM adds a time dimension to the 3D model, allowing for real-time simulation of construction progress. The 5D-BIM, on the other hand, adds a cost dimension to the 3D model, enabling the instant generation of cost budgets and financial representations of the model over time [19]. 5D-BIM can be created either by adding cost information to the 3D model objects and components or through a live connection to estimation software tools [20,53]. The 5D-BIM enables users to estimate costs, create cost baselines, visualize and track costs over the project life cycle, and evaluate different construction methods and alternatives [20].

*2.5. Cost Management*

Cost management is a critical component of project success, as it aims to minimize the cost of the project while maintaining acceptable levels of quality and scope. This process provides value for money for the client and ensures that the contract amount remains within the authorized budget or cost limitations [2,54].

In the past, cost management was reactive to changes in project scope, but now there is a shift towards incorporating it as a strategic aspect [55]. The construction industry is heavily influenced by professional standards and bodies such as the International Cost Engineering Council (ICEC), the Royal Institution of Chartered Surveyors (RICS), the Association for the Advancement of Cost Engineering International (AACEI), and the International Federation of Surveyors (FIG). These organizations provide expertise in cost engineering and management through professional standards that codify best practices and align understanding across the industry [56,57].

The AACEI defines cost management as a systematic approach to managing cost throughout the life cycle of any enterprise, program, facility, project, product, or service. The AACEI total cost management (TCM) Framework provides a hierarchical structure for best practices in the industry [58]. The RICS introduced the new rules of measurement (NRM) suite in 2009, which serves as a comprehensive reference for cost management in construction projects [59]. The International Cost Management Standard (ICMS) was developed in 2017 and has been revised to incorporate life-cycle costing and environmental sustainability in 2019 and 2021, respectively [60].

Cost management in construction has been traditionally approached with financial measures only, but this approach has been criticized for its limitations. Scholars have pointed out issues such as lacking metrics [61], failure to identify performance problems [1,62], lack of strategic focus, and hindrance to continuous improvement [63].

As a result, alternative approaches to cost management have gained popularity, including key performance indicators (KPIs) [64,65], benchmarking [66–68], and BIM [69,70]. KPIs, initially proposed by Cox et al. [71], reflect the quality of project outputs and outcomes and are used for performance evaluation. However, the development of KPIs for mega-projects has yet to reach sufficient levels, and excessive development can be a waste of time and resources [2].

Benchmarking, which involves comparing project processes, practices, and operations to similar projects, aims to identify strengths and weaknesses [68] and find the best practices to implement for improved performance [72]. Despite its benefits, benchmarking can

be criticized for lacking objectivity and neglecting intangible factors that impact project performance [73].

Traditional cost management approaches, such as bill of quantities (BoQ) or resource-based costing (RBC) [74], have also been criticized for their uncertain information and arbitrary allocation of overheads [75]. Activity-based costing (ABC) was developed to address these limitations and accurately allocate project overheads based on cost drivers [76,77].

Another promising approach in construction is target value design (TVD) [78,79], which reverses the traditional practice of cost estimation by having cost and value inform design decisions. Derived from target costing, TVD has roots in the manufacturing industry [80] and is implemented in construction through practices such as design–build–own–transfer, public–private partnership, integrated form of agreement, and integrated project delivery. However, successful TVD implementation requires a collaborative effort, and its application in less collaborative project delivery arrangements may lead to unintended consequences.

Cost management includes four major functions: cost estimation; cost modelling; cost budgeting; and cost/budget monitoring and control [81], each of which are outlined in more detail in the sections below.

### 2.5.1. Cost Estimation

Cost estimation involves the calculation and prediction of the time, cost, and other resources required to meet the project objectives [82]. The accuracy of cost estimation is influenced by the information related to the project's structure and characteristics [83]. Accurate cost estimation is crucial for making informed decisions and project success [84]. Cost estimates should always be presented with a plus/minus percentage, depending on the project scope definition [85]. Despite the significant investments in infrastructure mega-projects, there is limited research on cost management strategies and methodologies in this domain. Current literature focuses on cost estimation on the project level [86,87]. Cost estimates can be conducted using top-down techniques such as analogy or parametric, or using bottom-up techniques [84].

Lovallo and Kahneman [88] proposed supplementing traditional forecasting methods with "reference class forecasting," an objective forecasting method that overcomes sources of optimism. However, this approach was criticized for underrating deliberate forecast fabrication as a source of bias [89].

### 2.5.2. Cost Modelling

A Cost model is a framework for calculating the overall project value, aggregating cost estimating details into a total cost estimate [90]. Cost estimating methodologies have been classified in various ways, including analogy-based, parametric, engineering [91], qualitative (intuitive and analogical), and quantitative (parametric and analytical) [92].

### 2.5.3. Cost Budgeting

Cost budgeting is the process of aggregating estimated costs of individual activities or work packages to create a cost baseline and allocate resources for executing different project activities [93]. This budget provides the basis for management to make decisions, plan, control, and govern the project [94]. The early cost estimates serve as the foundation for the project budget. Before it can be termed a budget, early cost estimates must go through a prescribed process of reviewing asset development plans, project screening, and resources commitment for future project development [85]. The cost estimation process consists of five steps: defining the estimate basis, developing a base estimate, assessing risk and setting contingencies, reviewing the total estimate, and conveying the estimate [95]. Contingency is the amount of funds required above the budget to reduce the risk of overruns to an acceptable level for the organization. A common risk management strategy is to have a contingency reserve for known unknowns and a management reserve for unknown unknowns [96].

Despite the use of various approaches to determine contingencies, including traditional percentage [97,98], Monte Carlo simulation [99–101], artificial neural networks [102], theory of constraints [103], and reference class forecasting [104], these methods were criticized for being inefficient when it comes to cost overruns [96,105].

### 2.5.4. Cost/Budget Monitoring and Control

Cost and budget monitoring and control are vital to deliver the project on time, within budget, and within scope. The process includes assessing project progress, comparing it to the plan, analyzing variances, and implementing corrective actions [106]. There is a debate on whether project performance should be measured against the budget and schedule (focused on addressing deviations from the project plan (difference between should and did) [107–109]) or the value delivered to the client [108]. Techniques for budget control include project management information systems (PMIS) [110], earned value management (EVM) [111,112], work breakdown structure (WBS) [113] and, as a recent addition, building information modelling (BIM) [114]. Effective cost management in mega-projects requires careful planning, with the WBS and cost breakdown structure (CBS) codes commonly combined to support financial decisions and budget [20,115].

## 3. Materials and Methods

### 3.1. Approach

The present study aims to investigate the complex interplay between various domains concerning the implementation of 5D-BIM for cost control and management in rail infrastructure projects. Given the complexity of the research problem and the need to synthesize existing knowledge from multiple sources, a systematic literature review (SLR) approach was adopted.

The SLR approach offers two key advantages: transparency and exhaustiveness. It enables other researchers to replicate the study and facilitates the identification, evaluation, and synthesis of existing literature on the topic. Moreover, it aims to communicate the known and unknown aspects of a topic and provide recommendations for future research [24]. The following section provides a brief description of the SLR methodology employed in this study.

The previous literature review studies on cost overruns and 5D-BIM implementation have used a range of research approaches. Vigneault et al. [19] conducted a systematic literature review and introduced an innovative 5D-BIM framework for construction cost management. Sepasgozar et al. [116] used a mixed method of bibliographic analysis and content review to identify different uses of BIM to improve cost management. Shishehgarkhaneh et al. [117] conducted a bibliometric and systematic literature review on the use of BIM and digital technologies in the construction industry. Meanwhile, publications on BIM in the rail sector cover a broad range of topics with intriguing recommendations for future research. The authors of [22] presented a case study on integrating BIM into rail projects, while [118] called for strategic BIM adoption from the Korean railroad public owner's perspective. This research focuses on the implementation of 5D-BIM in rail mega-projects.

### 3.1.1. Systematic Literature Review Stages

Following an initial desktop study and literature exploration, a rigorous protocol and search strategy were developed based on the observations. According to Moher [119], "The preparation of a protocol is an essential component of the systematic review process; it ensured that a systematic review is carefully planned and that what is planned is explicitly documented before the review starts, thus promoting consistent conduct by the review team, accountability, research integrity, and transparency of the eventual completed review". The protocol outlined the rationale and planned methods for the review, including the defined SLR boundaries, by identifying inclusion and exclusion criteria, focusing on the period

between 2000 and 2023. The SLR followed the five-stage review approach as described by Pawson et al. [120] (see Figure 2).

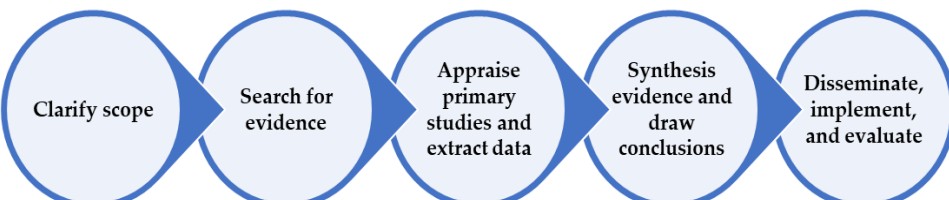

**Figure 2.** Systematic literature review stages.

The stages included: (1) clarifying the research question(s); (2) searching for relevant literature; (3) selecting relevant studies; (4) appraising the quality of the selected studies; and (5) synthesizing the results of the selected studies.

To report the SLR findings, the preferred reporting of items for systematic reviews and meta-analyses (PRISMA) [121] guidelines and flow chart were used. PRISMA is an evidence-based minimum set of items for reporting systematic reviews and meta-analyses, which ensures the completeness and transparency of the reporting process.

### 3.1.2. Tools and Software Packages

A combination of software packages was used for data collection, processing and exporting, as shown in Table 2 below.

**Table 2.** Tools and Software packages.

| Software Package/Tool | Utilization | References |
|---|---|---|
| VOS viewer | SLR data visualization and analysis. | [122] |
| Covidence | References screening, filtering, tagging and blind review. | [123] |
| CiteSpace | Analysing SLR clusters/trends and patterns. | [124] |
| EndNote and Mendeley | Manage/share/sort references library throughout the SLR process. | [125,126] |
| Microsoft Excel | Data collection, storage and visualisation. | [127] |

### 3.1.3. Data Sources

As shown in Table 3, The following electronic databases were used for data collection. The search algorithm for Google scholar is not known and cannot be controlled; Google adapts the search to each user in order to personalize information and, as a result, a systematic search is quite probably not replicable [128]. Consequently, Google Scholar was considered as an additional source only for this SLR.

**Table 3.** Search databases.

| Main Sources | | |
|---|---|---|
| 1 | Scopus | https://www.scopus.com (accessed on 8 January 2023) |
| 2 | Science Direct | https://www.sciencedirect.com (accessed on 8 January 2023) |
| 3 | Web of Science (new website) | https://www.webofscience.com (accessed on 8 January 2023) |
| Additional sources | | |
| 4 | Google Scholar | https://scholar.google.com (accessed on 8 January 2023) |

The PRISMA flow chart in Figure 3 summarizes the initial search process, which resulted in a total of 4342 papers from four databases: Scopus, Science Direct, Web of Science, and Google Scholar. The search results were verified through external access provided by researchers from Chalmers University of Technology and Northumbria University, and were then imported to EndNote and exported to Covidence in RIS format.

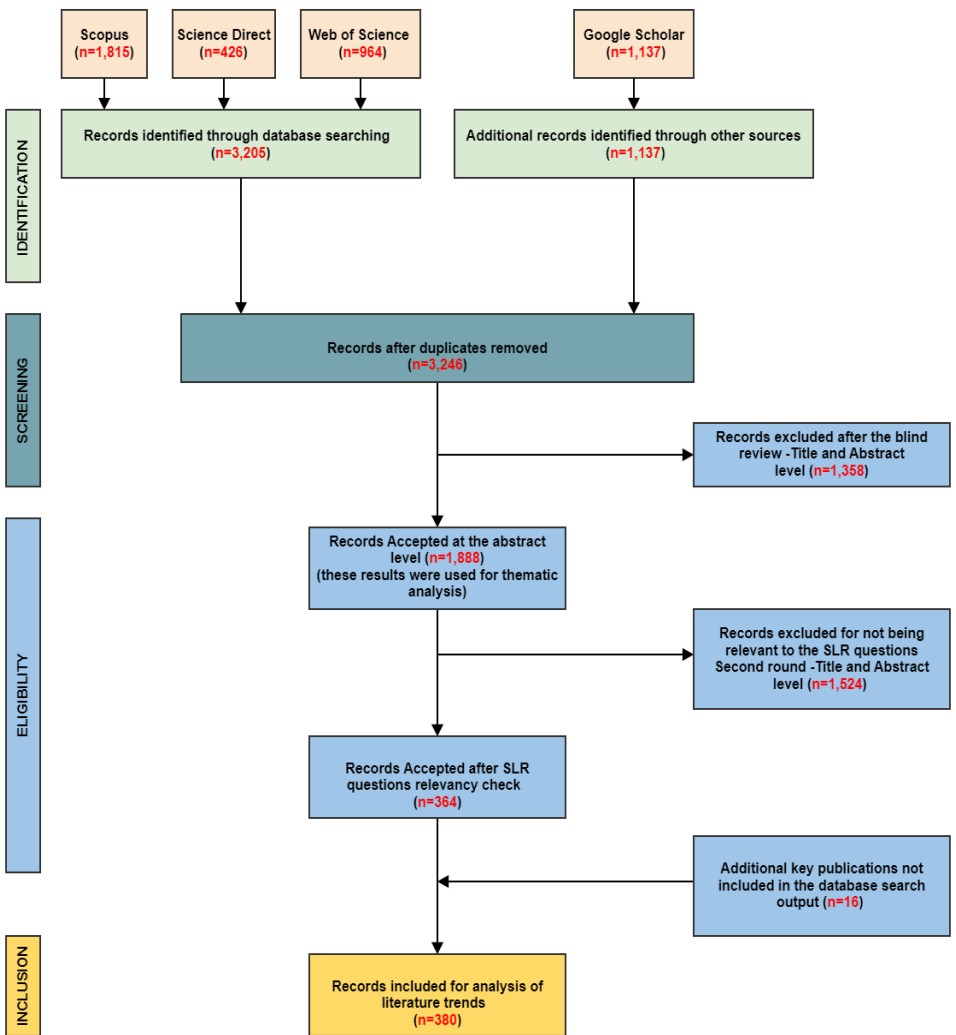

**Figure 3.** PRISMA flow diagram for the study.

After removing duplicates using Covidence, a blind review was conducted by the author and another team member, resulting in the identification and resolution of 270 conflicts.

The blind review considered the inclusion and exclusion criteria outlined in Appendix F, as well as relevance to the research domain. Despite the application of search filters, a considerable number of publications originating from the healthcare and medical domain surfaced, where the acronym BIM denotes a protein named Bcl-2-interacting mediator of cell death. Therefore, the blind review resulted in the exclusion of 1358 results, with 1888 results remaining.

Five themes surfaced during the initial review of the SLR data: BIM for construction, BIM for infrastructure, BIM for rail, cost management and control, and rail. These themes were used to tag and categorize data in Covidence.

Of the 1888 publications, 896 (47%) were tagged as "Cost Management and Control" by blind reviewers. This indicates a significant amount of coverage in the literature, with 611 journal papers, 259 conference papers, 18 books/book chapters, and 8 theses. However, the tag combination of "BIM for Infrastructure," "BIM for Rail," "Cost Management and

Control," and "Rail" yielded only 35 results (20 journal papers, 12 conference papers, and 3 theses), suggesting that this sector is underrepresented in the literature.

The 1888 results remained after the initial blind review were used for thematic analysis to identify research trends and gaps. A second screening round using the same blind review technique was conducted in order to address the four research questions. During this stage, publications discussing topics such as cost overruns in transportation and rail projects, cost models, various cost management and control strategies, and 5D-BIM were included for full-text review. As a result of the second screening stage, 364 publications were retained for further analysis. Additionally, sixteen publications that were not captured in the search were added to the review process.

The final number of records included in the analysis was 380 publications. To aid in further analysis, a thesaurus file was created and uploaded to VOS viewer, allowing for the combination of synonyms such as BIM, Building Information Modelling, and Building Information Modelling under a single term. Similarly, terms such as Cost Overrun, Cost Escalation, and Budget Overrun were combined under the term Cost Overruns. Finally, CiteSpace was used to analyze clusters, trends, and patterns among the results.

### 3.2. Network Representation

The application of network theory was utilized to understand the structure of the literature. This structure can be represented by two key components: network actors and network ties. Keywords were used to describe the contents and discussion topics of each study [129]. To map the occurrence of keywords, a network representation was created using VOS viewer. In this representation, colors indicate knowledge areas while node size represents the number of occurrences in the literature; a larger node indicates a larger knowledge area. The links between nodes represent the citations in pair and group articles, and these links become stronger (closer nodes) when two neighboring nodes have similar co-authors or frequent pair citations [122].

The network analysis in this study was conducted in two stages. In the first stage, networks were created using VOS Viewer by analyzing keyword co-occurrence and co-authorship. In the second stage, maps were generated using CiteSpace to extract useful information, make sense of the generated networks, and reveal trends and patterns.

### 4. Results

#### 4.1. Publication Sources

As shown in Table 4, out of 1888 result 1277 were journal articles (67.6%) and 565 conference papers (30%). Other sources included in the study are: 23 books (1.2%) and 23 thesis (1.2%). It was noticed from the analysis that the number of conferences papers covering rail and BIM exceeds the journal articles, while cost overrun is covered mostly by journal articles.

**Table 4.** Distribution of tags among publications based on 1888 records.

| Tag | No | Conference Paper | Journal Paper | Book | Thesis |
|---|---|---|---|---|---|
| BIM for Construction | 136 | 34 | 100 | 1 | 1 |
| BIM for Construction; Cost Management and Control | 244 | 76 | 165 | 0 | 3 |
| BIM for Infrastructure | 51 | 19 | 31 | 0 | 1 |
| BIM for Infrastructure; BIM for Rail; Cost Management and Control; Rail | 35 | 12 | 20 | 0 | 3 |
| BIM for Infrastructure; BIM for Rail; Rail | 191 | 67 | 118 | 2 | 4 |
| BIM for Infrastructure; Cost Management and Control | 55 | 18 | 37 | 0 | 0 |
| Cost Management and Control | 896 | 259 | 611 | 18 | 8 |

**Table 4.** *Cont.*

| Tag | No | Conference Paper | Journal Paper | Book | Thesis |
|---|---|---|---|---|---|
| Cost Management and Control; Rail | 180 | 51 | 127 | 0 | 2 |
| Rail | 100 | 29 | 68 | 2 | 1 |
| Total | 1888 | 565 | 1277 | 23 | 23 |

*4.2. Analysis of Publication Source*

Figure 4 illustrates the yearly count of BIM publications. Although the trend indicates a general increase in BIM publications over time, there were two noticeable drops in the years 2007 and 2015, which is supported by both BIM literature [130,131] and industry reports [132]. Such trend aligns with the typical life cycle of technology [133], which suggests a shift from research and development towards wider adoption. In 2016, the UK government mandated the use of BIM in the public sector, which triggered a new cycle and led to increased industry funding and a subsequent surge in BIM publications [130].

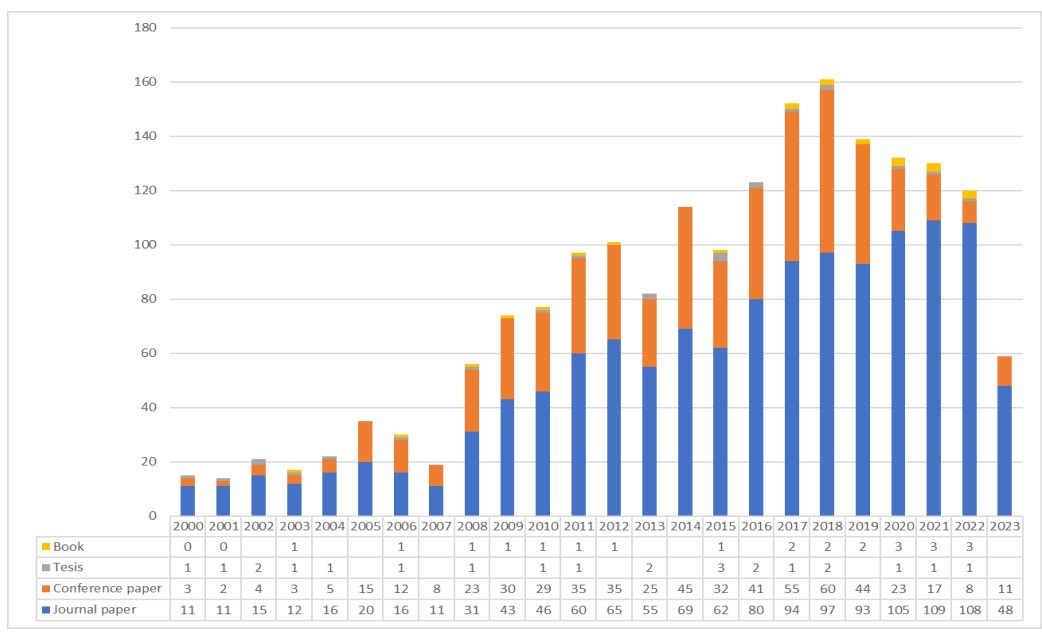

**Figure 4.** Distribution of publications per year (as of February 2023).

The drop in BIM publications during the years 2020, 2021, and 2022 may be due to the impact of COVID-19. The pandemic caused significant setbacks to scientific research worldwide, as travel, social, and funding restrictions took their toll. As a result, research personnel and resources were intentionally directed towards COVID-19 activities, above all other endeavours [134].

In 2023, as life returns to normal, a significant increase in BIM publications can be observed. In the first two months of the year, the number of BIM publications has already reached 48.

As shown in Figure 5, the highest number of relevant publications came from two journals: *Automation in Construction*, and *Journal of Construction Engineering and Management*. BIM can be used to automate various construction processes, such as design, cost estimation, and scheduling. As a result, the *Automation in Construction* journal, which focuses on advancements in construction automation and decision support systems, attracts a significant number of publications related to BIM.

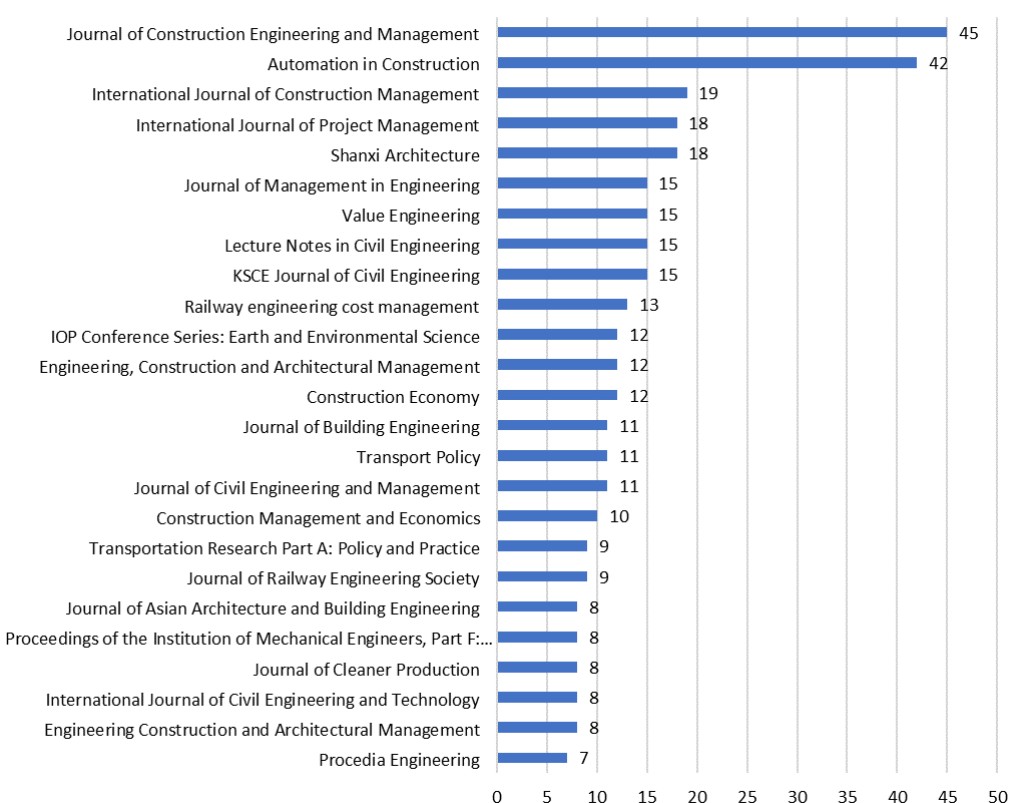

**Figure 5.** Number of publications and source.

Likewise, research articles related to BIM are highly relevant to the *Journal of Construction Engineering and Management*, which focuses on advancements in the theory and practice of construction engineering and management.

Other active journals in the field include *Shanxi Architecture, KSCE Journal of Civil Engineering*, and the *Journal of Railway Engineering Cost Management*.

*4.3. Location Analysis*

Analysing the distribution of publications by country is crucial to understand the current state of academic research and knowledge production worldwide. By examining which countries are publishing the most, as well as which disciplines and topics are being researched, researchers and policymakers can gain insights into the strengths and weaknesses of various national research systems, identify disparities in research funding, as well as highlighting areas where greater collaboration and knowledge sharing might be beneficial [135,136].

The wrapper data mining platform [137] was utilized to produce a publication analysis based on the lead author's country of origin, using the data related to their country. Figure 6 shows the distribution of publications by country of lead author; top contributors include the United States, the United Kingdom, and China. Australia, Germany, Italy are following the lead. The results show a noticeable surge in BIM publications published by authors from China. This could be attributed to a combination of government support, rapid urbanization, technological advancement, and growing industry, which overall created a strong demand for BIM [138,139].

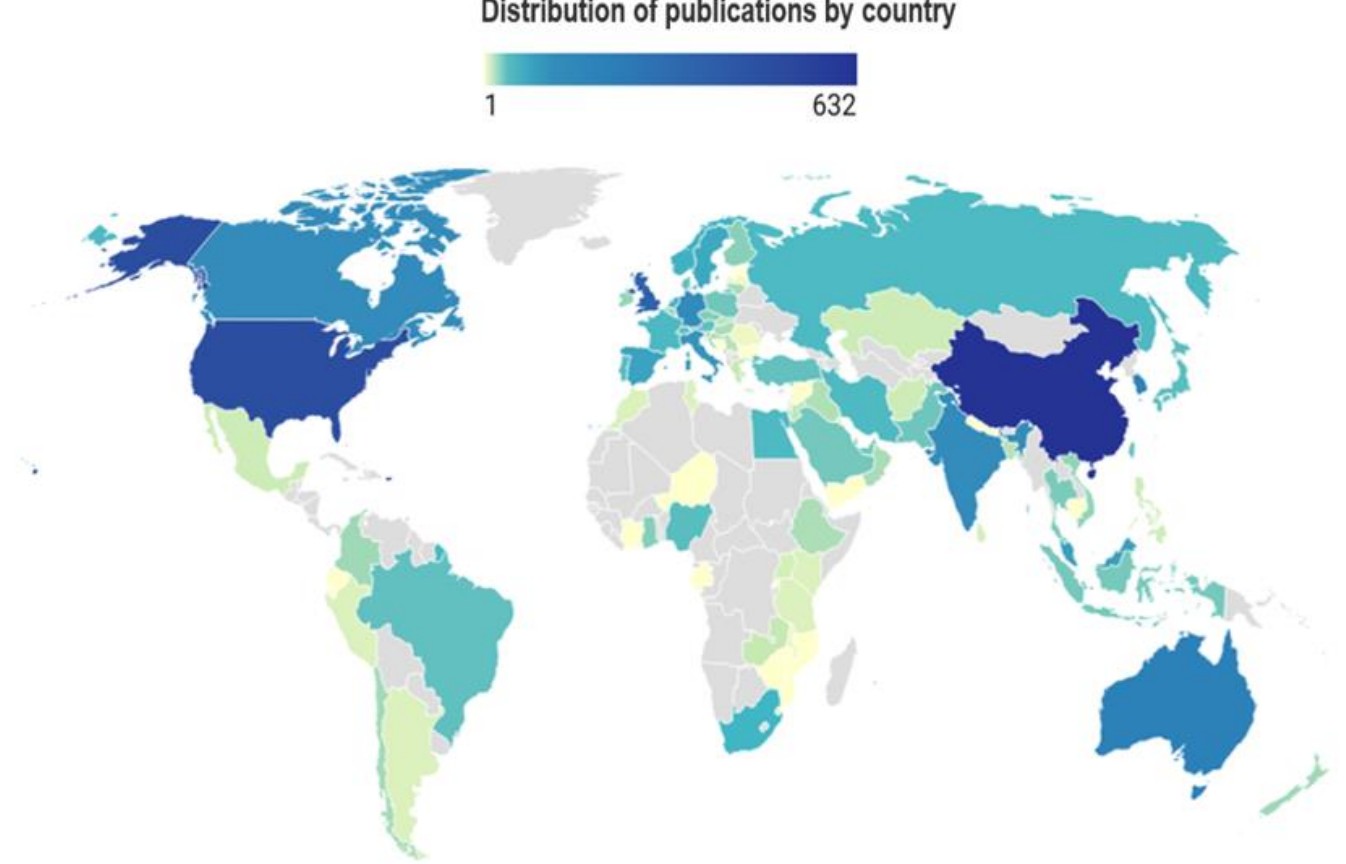

**Figure 6.** Distribution of publications by country.

*4.4. Co-Authorship Network*

Knowledge exchange, idea sharing, and creativity are all facilitated by collaborations between or among researchers [140]. These collaborations are also effective in joint funding applications [141]. Co-authorship and citation are good indexes for research productivity and synergy between different knowledge areas, thereby a co-authorship network was created to highlight the links between active authors.

A co-authorship network is a visual representation of the collaboration patterns among authors of academic publications in a specific field or topic. The nodes represent the authors, and the links represent (denote) the coloration through co-authorship. A minimum of two publications was considered to create the network.

Fifteen co-authorship networks came out as significantly relating to themes and expertise; Figure 7 and Table 5 shows different research groups/authors and their area of contribution.

**Table 5.** Different research groups/authors and their area of contribution.

| Group | Research Theme | Key Authors/References |
|:---:|:---|:---|
| 1 | Cost overrun | [9,40,42,105,142–147] |
| 2 | Cost causation | [148–151] |
| 3 | BIM implementation impact on project cost management | [47,152–154] |
| 4 | Integrating BIM into railway projects | [22,155–157] |
| 5 | BIM utilization in railway design | [158–164] |
| 6 | Cost overrun typology in infrastructure projects | [165–168] |
| 7 | Smart railway systems | [169,170] |

**Table 5.** *Cont.*

| Group | Research Theme | Key Authors/References |
|:---:|:---|:---:|
| 8 | Cost modelling, risk and contingency calculations | [171–180] |
| 9 | Cost overrun impact assessment | [181] |
| 10 | Target costing process and design | [182–185] |
| 11 | Railway lifecycle costing | [186–190] |
| 12 | Project performance and cost control | [191–197] |
| 13 | BIM and sustainability | [198] |
| 14 | BIM implementation analysis | [152,153,199–205] |
| 15 | Utilizing BIM and immerging technologies in railway industry | [206–208] |

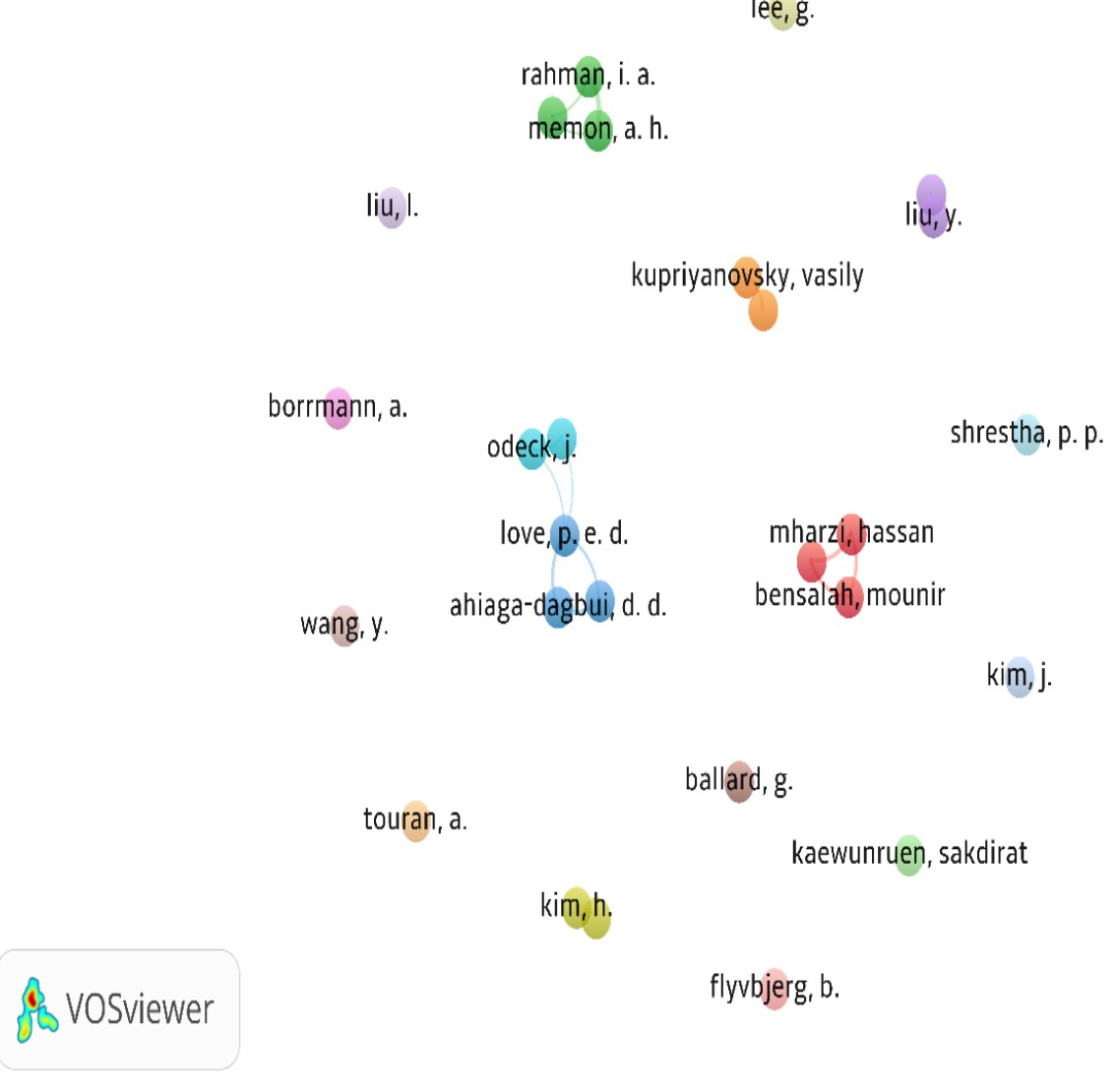

**Figure 7.** Network representation of authors' citations in literature.

*4.5. Keywords Re-Occurrence and Cluster Analysis*

In Figure 8, the SLR results are presented as five main clusters: Cost overrun, BIM, life cycle, Railway, and Cost estimation, each comprising closely related themes. The BIM cluster, for example, includes themes such as buildings, BIM adoption and implementation, and quantity surveying.

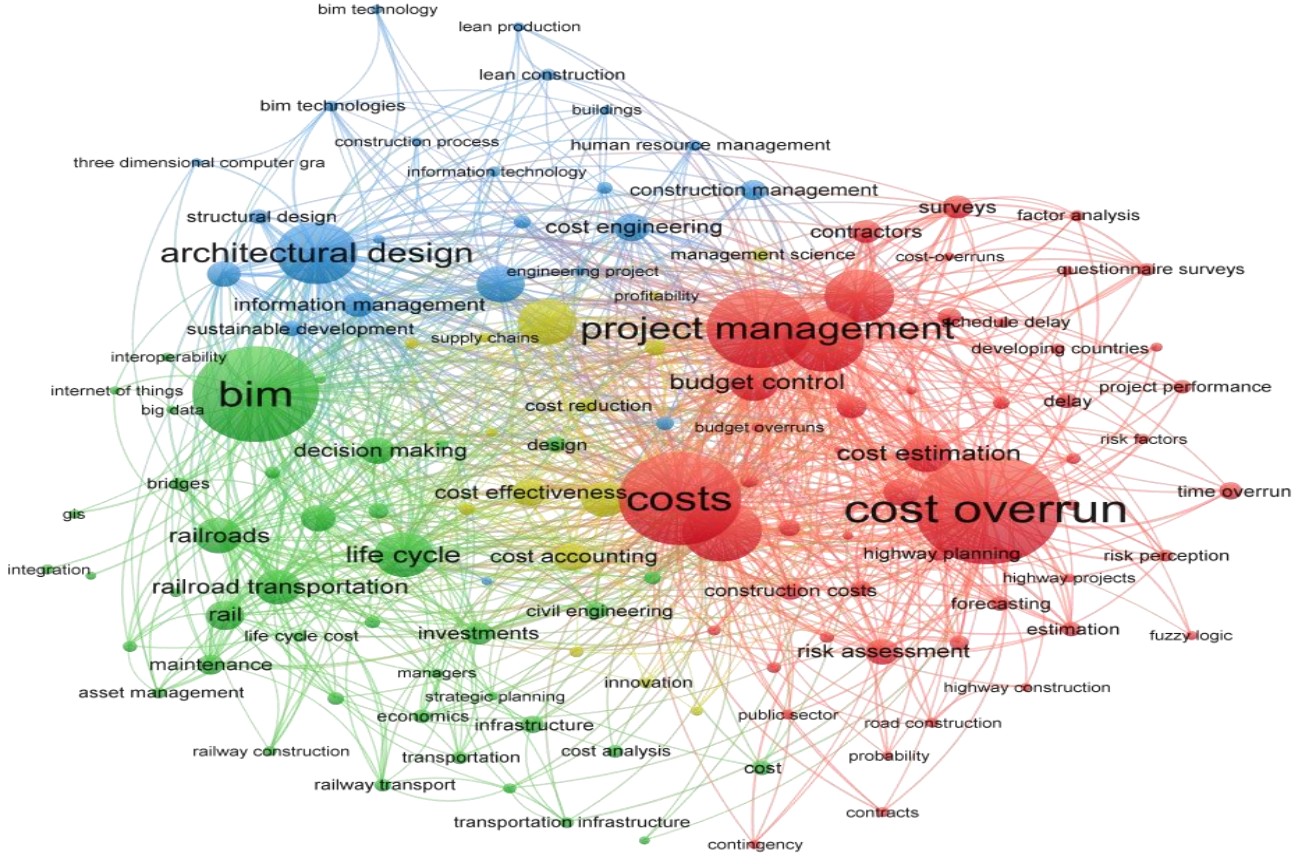

**Figure 8.** Network representation of keywords re-occurrence in the literature based on 1888 records.

## 5. Findings and Discussion

VOS Viewer has limited cluster analysis capabilities, so CiteSpace was employed for a more comprehensive analysis of clusters. CiteSpace can automatically organize nodes into clusters, which can uncover the underlying class structure of a network [209,210].

After filtering small clusters, nine main clusters were identified using CiteSpace, as shown in Figure 9 and Table 6. These clusters have a modularity value (Q) of 0.45 and silhouette ranging from 0.61 to 0.94, indicating reasonable intra-cluster similarity.

**Table 6.** Key network clusters.

| Cluster ID | Size | Silhouette | Label (LLR) | Average Year |
|---|---|---|---|---|
| 0 | 179 | 0.628 | Railway infrastructure | 2011 |
| 1 | 155 | 0.617 | Building Information Modeling | 2014 |
| 2 | 153 | 0.668 | Construction project | 2008 |
| 3 | 107 | 0.609 | Demand forecast | 2015 |
| 4 | 95 | 0.830 | Cost-effectiveness analysis | 2009 |
| 5 | 75 | 0.744 | Supply chain | 2008 |
| 6 | 67 | 0.817 | Cost deviation | 2013 |
| 7 | 51 | 0.820 | Construction contract | 2008 |
| 8 | 32 | 0.882 | Cost control | 2012 |
| 9 | 17 | 0.943 | Troubled project | 2007 |

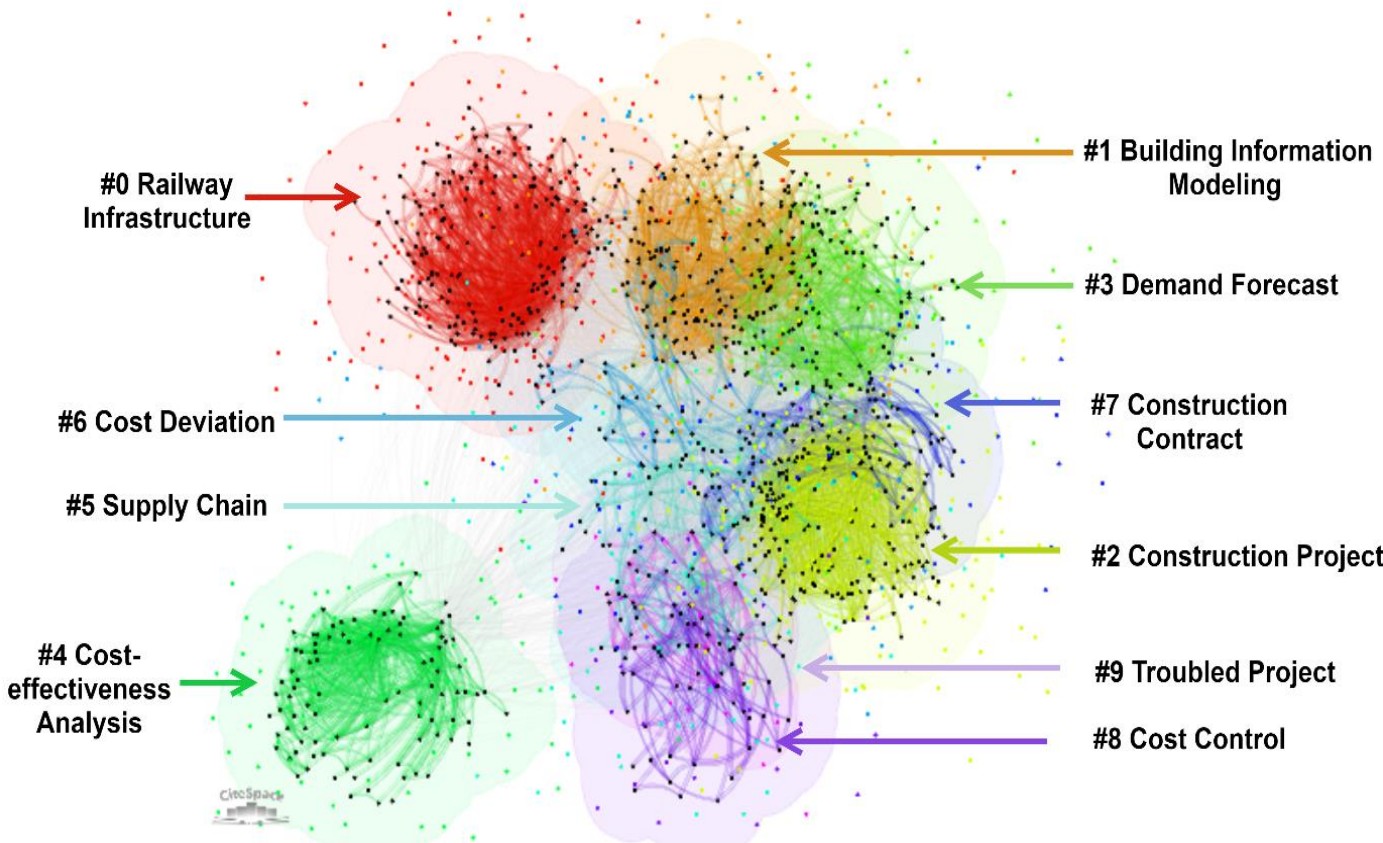

**Figure 9.** Key network clusters.

### 5.1. Cluster # 0 Railway Infrastructure

The largest cluster (#0) has 179 publications and a silhouette value of 0.628. This cluster is mainly associated with economic analysis and life cycle costing for rail projects. The major five citing articles of the cluster are [211–215]

The most cited words in this cluster are: 253 life cycle,172 railroad, and 144 railroad transportation.

Conceptual Debates in Cluster # 0

In this cluster we find deep conceptual debate about the directionality of cost overrun, where the definitions come from, and different perspectives.

The cluster discusses the issue of cost overrun in rail projects which continues to be a persistent challenge despite efforts to ensure cost-effective delivery. Studies and government reports have highlighted the prevalence of cost overruns in infrastructure projects, including rail [3,6]. According to Flyvbjerg [144], rail projects have the highest mean cost overrun compared to other transportation projects at 44.7 percent. However, the reported mean varies among studies due to differences in how cost overrun is defined and measured. Canteralli et al. [216] define cost overrun as the difference between initial forecasted budget and actual construction costs, while Odeck [42] proposes that the reference point should be at the detailed planning stage where the final cost is determined. Love et al. [6] suggest that cost overrun should be measured from the point of construction contract signature. However, traditional research methods and designs have resulted in misleading conclusions regarding cost overrun. Therefore, developing a robust theoretical frameworks to understand and predict cost overrun is crucial to mitigate its occurrence in rail projects (see Love et al. [105]).

When it comes to the relationship between BIM and rail projects, this cluster highlights that the use of BIM in infrastructure, particularly in rail projects, is gaining momentum. While BIM is commonly used in the Architecture, Engineering, Construction, and Operations (AECO) industry, its implementation in infrastructure is three years behind [217]. However, recent literature shows an increase in its use. Although the concepts of BIM in AECO and rail are the same, the key advantage of BIM in AECO, such as visual aid, is less significant in linear projects such as rail. The rail project lifecycle has five stages: planning, survey, design, construction, and operation. Figure 10 illustrates the benefits of BIM at each stage of the rail project lifecycle.

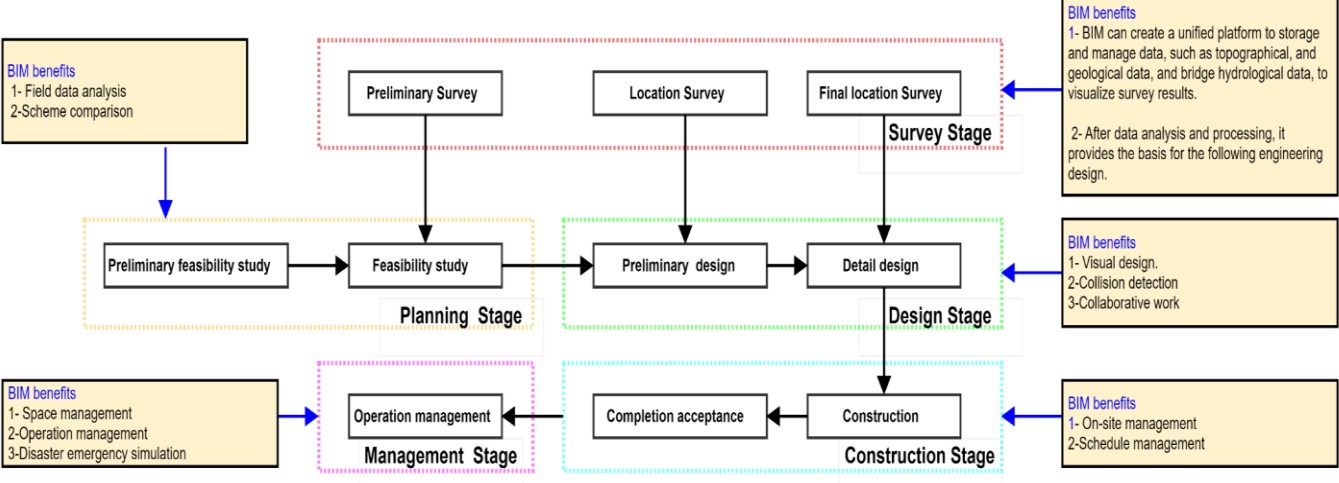

**Figure 10.** Schematic diagram on the makeup of railway engineering, adopted from [218]. Copyright 2018 by ASCE. Reprinted with permission.

Although the benefits of BIM implementation in rail projects are evident, many companies are facing significant challenges in its implementation [219,220].

These challenges include complex network topology, high environmental requirements, compliance with international and national standards, and the need to meet various supply divisions and track requirements [221,222].

Bawono et al. [221] categorized these challenges as technical, personal, and process-related, as per Table 7.

**Table 7.** BIM implementation challenges in rail projects [221]. Copyright 2021 by Springer Nature Springer Nature. Reprinted with permission.

| Issues | Challenges |
|--------|-----------|
| Technical | Handling of growing file sizes |
| | Lack of standardized data exchange |
| | Lack of proper design software |
| Personal | Change attitude and mindset of people |
| | Motivate people |
| | To have the same understanding of BIM within the whole project team |
| Process | Transition from 2D to 3D design characteristics |
| | Absence of standardized data exchange |
| | BIM only works if the client is completely convinced to use BIM |

*5.2. Cluster # 1 Building Information Modelling*

The second largest cluster (#1) has 155 publications and a silhouette value of 0.62. This cluster refers to BIM benefits, different BIM uses in construction industry, and the evolution of GIS, the Internet of Things (IoT), and BIM application in the rail industry. The major five citing articles of the cluster are [223–227]. The most cited words in this cluster are: 285 architectural design, 164 BIM, and 157 construction.

This cluster discuss the integration of BIM into rail projects which continues to be a growing trend worldwide [228], with the majority of publications on rail infrastructure coming from the United Kingdom and China [229]. Deployment of the BIM process in railways is expected to continue between 2020 and 2030, and the benefits of BIM have been demonstrated in railway station construction projects, rail track rehabilitation, stakeholder management, and decision-making processes. However, despite the efforts to develop a 5D-BIM system that combines cost, schedule, and a 3D-BIM model, few have been successful due to its complexity. Once developed, the advantages of a 5D-BIM system are numerous, including real-time visualization and verification of the cost and schedule, as well as forecasting the future cost and schedule [21,188,219,227,230].

This cluster also looks at BIM and its potential to mitigate cost overruns, which has been a topic of interest in the literature. 5D-BIM, which combines cost, schedule, and 3D-BIM models, has been proposed as a solution, but its complex technology has hindered successful implementation. However, if developed, 5D-BIM has the potential to provide real-time visualization and verification of cost and schedule, as well as the ability to forecast future costs and schedules [231]. Through a thorough literature review, a correlation between the causes of cost overruns and the benefits of BIM was identified and is presented in Table 8.

**Table 8.** Key cost overrun causes from the literature and corresponding BIM advantage.

| Key Cost Overrun Causes | Corresponding BIM Advantage |
| --- | --- |
| Poor planning [6,232] | Improved planning processes [233,234]. |
| Strategic misrepresentation, i.e., lying [40]. | Transparency in decision making and data sharing [234,235] |
| Forecasting errors including price rises, poor project design, and incompleteness of estimations [236–238]. | Improved cost management for design stage [19,239,240] |
| Scope changes [241]. | Improved scope control [240]. |
| Poor cost estimation [242]. | Improved cost estimation processes [19,239]. |
| Frequent design change during construction phase [242]. | Improved change management during design process [243,244]. |

Another topic which is discussed in this cluster is the different uses of 5D-BIM cost management in the rail industry. The current 5D-BIM uses in the industry include quantity take-off, cost estimation, cost budgeting, cost control and lifecycle cost analysis.

5.2.1. Quantity Take-Off (Quantification)

Building and infrastructure projects require quantification to estimate their costs, but traditional methods are time-consuming and prone to human error and poor coordination of information [245]. Using BIM models for quantity take-off allows for faster and more efficient production of materials schedules, as discussed by Gaur and Tawalare [227], Aibinu and Venkatesh [246], and Stanley et al. [247]. However certain quantities cannot be extracted directly from the models due to the current tool's capabilities, the information structure in the model, or simply because some elements are not modelled. Cost engineers /quantity surveyors will need a mix of traditional methods/experience to model missing information [248,249].

### 5.2.2. Cost Estimation

5D-BIM allow more precise cost estimates and overall costs reduction [249]. It also allows designers to be aware of the cost effects/consequences of their changes early enough to help curb excessive budget overruns caused by modifications [250].

The quantity take off tools in 5D-BIM can be used to generate accurate project estimates [243]. Employing 5D-BIM tools could be deceptive; instead, it is preferable to export 5D-BIM metadata, process it externally, and use the outcome for more realistic estimates [251,252].

Developing cost estimation practice guidelines and sharing the benefits of 5D-BIM estimation techniques with clients is crucial for the effective use of 5D-BIM in cost estimation [253].

### 5.2.3. Cost Monitoring and Control

Cost monitoring and control through the project lifecycle can be improved by adopting 5D-BIM; the positive impacts include: automated cash flow forecast and progress payments [254], direct procurement for different 5D-BIM elements [47], better change orders management [115,255], and better claim management [256].

### 5.2.4. Lifecycle Cost Analysis

Life-cycle cost analysis (LCCA) is one of the important tools to support decision making in infrastructure and rail projects. The strict budget limits and increasing performance and reliability requirements led infrastructure managers to develop computer-based tools (such as BIM) to better manage LCCA [257].

A LCCA's aim is to assess the overall costs of project alternatives and choose the design that assures the asset has the lowest overall cost of ownership compatible with its quality and function [258].

The LCCA should be undertaken early in the design process when there is still the possibility to modify the design to ensure a saving in life-cycle costs (LCC).

Lu et al. [259] emphasized the importance of using BIM for LCCA; they conducted a critical review for the integration of LCCA and LCC using BIM and introduced a framework for BIM-integrated LCCA and LCC.

Bensalah et al. [260] simulated an LCCA for a rail project (tram) using BIM; the analysis revealed that BIM would reduce 8.4% of the overall cost of the project, as well as 10% of maintenance costs over a 30-year period.

Zhao and Tang [261] focused on developing a full life-cycle cost management system module based on BIM. This module included a cost management platform and cost application software, which aimed to improve cost engineers' productivity, accuracy, and change management capabilities.

Despite 5D-BIM benefits and potential, overall development and the boundary of 5D-BIM is unclear and still at the early stage of adoption [20]. Appendix A shows different 5D-BIM uses as discussed in the literature.

### 5.3. Cluster # 4 Cost-Effectiveness Analysis, Cluster # 6 Cost Deviation, and Cluster # 8 Cost Control

The fourth, the sixth, and eighth largest clusters (#4, #6 and #8) have 252 publications and an average silhouette value of 0.7.

The main focus of clusters #4, #6 and #8 is on: the cost overrun phenomenon in transport projects, its definition, causes, and impact; and risk management and contingency calculation.

The major citing articles of the cluster are [4,8,9,94,262–265]. The most cited words in this cluster are: 595 cost overrun, 503 cost, and 345 project management.

A number of studies in these clusters have examined the landscape of cost overruns in this area [3–7]. Factors that contribute to cost overruns, including ambiguous project design and lack of coordination for design decisions among stakeholders, result in scope creep and rightful critique of unrealistic cost targets at the budget sign off [6,236,266]. Cost overruns can be grouped into several major categories, such as changes in project scope,

construction delays, and unreasonable cost estimation. Some studies have also explored the use of earned value management systems to prevent cost overruns [116].

However, there is debate about the root causes of cost overruns in transport infrastructure projects. Flyvbjerg et al. [40] suggested that strategic misrepresentation (i.e., lying) is the primary cause of cost underestimation in public works projects. Love and Ahiaga-Dagbui [8] challenged this claim, arguing that it is based on supposition and misinformation. They introduced two schools of thought to examine the phenomenon: evolutionists, who attribute overruns to changes in project scope, and psycho-strategists, who see deception, planning fallacy, and unjustifiable optimism as the primary causes of cost overruns.

Love et al. [105] refuted Flyvbjerg's suggestion that misrepresentation and optimism bias are the primary causes of cost overruns, arguing that these explanations ignore the complex array of conditions and variables that interact during project procurement. Flyvbjerg [267] accused Love et al. [105] of ignoring the basics of behavioral science and stressed that planners and managers consistently underestimate complexity and scope changes in projects. Several studies have attempted to use statistical measures of correlations between variables [268–270], but these have failed to provide convenient explanations, as correlation does not necessarily imply causation [6].

Cost overrun causes can be dependent on the viewpoint, and auditors often explain cost overruns as technical challenges with forecasting and delivering infrastructure [271]. The economic literature focuses on the perspective of the public decision-maker, while construction engineering managerial analysis focuses on contractual incompetence and related technical consequences [5]. Appendix C provides a list of key papers on cost overruns in transport and rail projects and a brief description of their contributions and conclusions.

The research collaboration on cost overrun and project lifecycle has focused on two main areas: cost escalation analysis and lifecycle perspective. The prevalent methodology in the cost overrun literature is to analyse the overall cost escalation between the project's early and final stages, with a particular emphasis on the execution stage.

Cavalieri et al. [272] analysed the cost overrun for transport projects, looking at how cost overrun changes over the different stages of the project life cycle. Government entities (contracting authorities) tend to overcommit to figures/numbers in the early stages when allocating budget, developing forecasts, and during budget approval stages. This can be linked to the current "optimism bias" and "risk aversion" outlined by Lovallo and Kahneman [88].

In the same context, Terrill et al. [273] examined the timing and magnitude of cost overrun in rail and road projects in Australia. The study revealed that poor compliance with project appraisal processes is correlated with a higher probability of cost overrun.

These clusters also discuss cost overrun prediction/estimation and analysis. The cost estimates during the early stages can harm the asset owner as well as the project team. Cost estimates affect project screening/budget approval, resource allocation, and further project development. In addition, one of the key performance assessment criteria for the project team's success is budget management and control; unrealistic early cost estimates lead to budget/cost overrun [274,275].

Different methods/techniques are used for cost overrun prediction, cost estimation, and cost contingency calculations in the literature [276]. Appendix E provides a summary for the most popular ones.

Cost Estimation Models

Another main discussion topic in these clusters is cost estimation models, which provide the basis for budget allocation and resource planning. Qualitative methods rely on the estimator's knowledge and experience to identify project characteristics and influencing factors. Quantitative methods use historical data analysis and quantitative techniques to estimate project costs [277]. Appendix B provides a summary of quantitative and qualitative cost estimation models.

Parametric cost estimating calculates project costs based on project parameters without considering minor details [278]. Analogous estimating uses similar past project data to estimate costs for new projects [279], while analytical methods provide detailed cost estimates for each project element or activity, resulting in a more accurate estimate.

Top-down estimates (analogous approach), which use historical project cost data to estimate the cost of the current project, are typically used in the conceptual phase of a project. In contrast, bottom-up estimates utilize the project work breakdown structure (WBS) and detailed information on each activity [280].

The choice of cost estimation method should be precise, accurate, and well-documented while remaining practical, easy to use, and cost-effective [281]. Various cost estimation models have been developed for rail and transport projects, including fuzzy expert systems, BIM, expert judgment, Monte Carlo simulation, use of historical data, case-based reasoning (CBR), unit cost, parametric, and artificial neural networks (ANNs).

For example, Byung Soo Kim [282] used case-based reasoning (CBR) with a genetic algorithm (GA) and multiple regression analysis (MRA) to design railway bridges. Shin et al. [283] confirmed the benefits of BIM in cost analysis in railway projects, and Barakchi et al. [284] found that parametric, artificial neural networks (ANNs), and Monte Carlo simulation are the most commonly used cost estimation models in rail projects.

Figure 11 provides a summary of the different cost estimation models used in the construction industry.

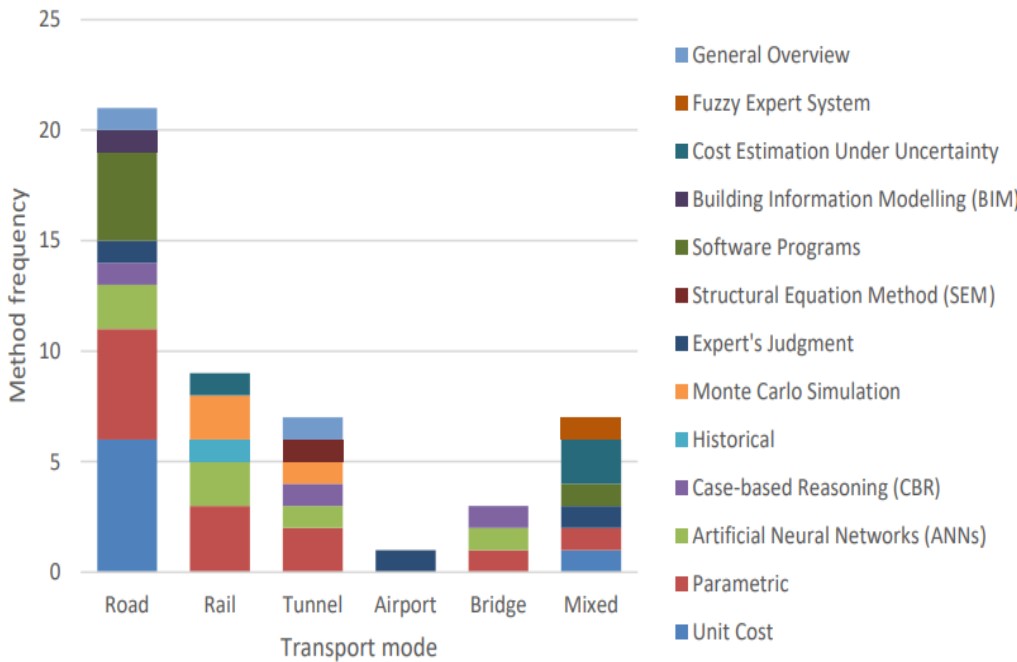

**Figure 11.** Cost estimation methods by each transport mode [284]. Copyright 2017 by Elsevier. Reprinted with permission.

Key publications covering cost models used for cost estimation, prediction, and analysis in transport and rail projects are shown in Appendix D. Another main discussion topic in this cluster is cost overrun mitigation/prevention.

Cost mitigation is a popular topic in the literature. Flyvbjerg [285] proposed using reference class forecasting (RCF) to overcome optimism bias and misrepresentation. RCF is based on planning and decision-making theories that received the Nobel Prize in Economics in 2002.

However, Love and Ahiaga-Dagbui [9] have identified limitations to RCF and caution that it can be misleading if an inappropriate distribution is used to determine uplifts. Another

strategy is better resource management and effective communication between a project's internal and external stakeholders, which Ahiaga-Dagbui et al. [242] have identified.

De Jong et al. [286] have suggested improving project estimates, mitigating project risks, promoting an accountability culture, and ensuring clear project scope and goals to avoid cost overruns in transport projects. Siemiatycki [271] recommends enhancing performance monitoring, reporting, and information sharing, accountability and responsibility for errors and overruns, management capabilities of staff, and applying state-of-the-art forecasting techniques. Additionally, historical project cost databases and data mining methodologies can create decision support systems that reduce cost overruns, according to Ahiaga-Dagbui and Smith [287].

Construction rework is another cause of cost overruns, as discussed by Love and Li [288], who recommend greater attention to quality management practices and implementation. Finally, Love et al. [8] recommend including a contingency in the final approved budget to accommodate possible cost overruns, although this approach has its own drawbacks. While these strategies are effective, they only target discrete elements of the project lifecycle, and their overall applicability is unclear.

### 5.4. Citation Burst and Trend Analysis

A citation burst shows that specific keywords appeared frequently in published studies over a specific time period, indicating activity on the topic and highlighting fast-growing areas of research [289].

Figure 12 shows that the fields of project management, mathematical modelling, and cost control are quite mature areas of research that received a lot of attention from 2010 to 2011.

| Keywords | Year | Strength | Begin | End | 2000 - 2023 |
|---|---|---|---|---|---|
| Project Management | 2000 | 24.33 | 2000 | 2011 | |
| Mathematical Model | 2000 | 11.29 | 2000 | 2010 | |
| Cost Control | 2000 | 20.51 | 2000 | 2008 | |
| Budget Overrun | 2005 | 12.73 | 2005 | 2013 | |
| Cost Accounting | 2005 | 18.11 | 2005 | 2012 | |
| Building Information Model (BIM) | 2010 | 12.58 | 2013 | 2019 | |
| Research | 2005 | 17.61 | 2009 | 2014 | |
| Lifecycle Cost | 2008 | 18.24 | 2009 | 2013 | |
| Civil Engineering | 2003 | 10.62 | 2009 | 2013 | |
| Value Engineering | 2010 | 10.57 | 2010 | 2014 | |

**Figure 12.** Top 10 keywords with the strongest citation bursts.

Beginning in 2013, BIM studies saw a surge in citation activity, which lasted until 2019, when new terms such as digital engineering emerged. Other terms related to cost management and control are included in the list because this field is mature and has constantly evolved with applications.

Although the extant literature on the topic is limited, the findings suggest an emerging tendency for artificial intelligence (AI) and machine learning (ML) publications in the railway domain [290]. Currently, rail projects are utilizing AI and ML to optimize the effectiveness and performance of railway systems by different means, including resources and equipment planning during the construction stage [291], delving into the causation factors of highway–rail crossing crashes [292], categorizing fatality rates for accidents [293], improving safety measures [294,295], mitigating collision risks [296], and integrating building information modeling (BIM) and ML to enhance the operation and maintenance of railway networks [297].

Figures 13 and 14 give a more detailed look at the outcomes. VOS Viewer automatically adjusted the contrast of the results, which revealed that prior to 2010, both *cost overrun* and

*economics* were mature areas of research that have been studied extensively over several decades and have developed a robust body of literature and empirical evidence.

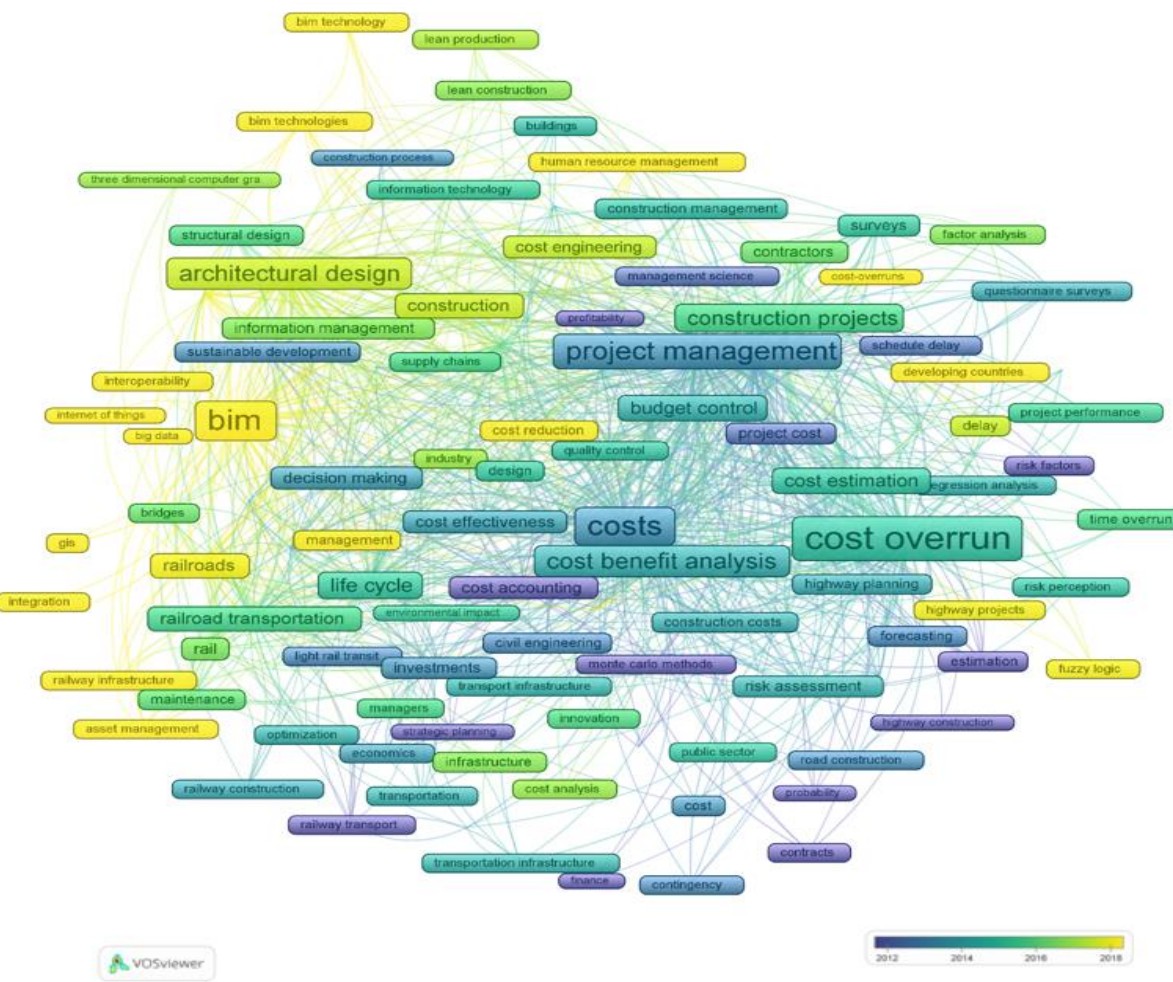

**Figure 13.** Time trend overlay visualization based on 1888 records.

Researchers have developed different *standards*, theoretical frameworks, models, and statistical methods to identify and predict *cost overruns* in different contexts. The research in this area has also led to the development of best practices and tools for managing cost overruns in various industries, such as construction, transportation, and defence.

During this period, *cost management* research primarily focused on *risk management*, *project cost*, *finance*, and *profitability*.

Similarly, *railway transport* is part of the broader *transport infrastructure research* and *public sector* projects domain, with increased focus/emphasis on *strategic planning*.

The results further demonstrate a clear shift in *cost management* research after 2015, with a growing focus on utilizing *information technology* and *computer-aided design* in **cost estimation**, *cost analysis*, *budget control*, and *life-cycle costing*. Concurrently, distinct research themes emerged within the *rail* industry, including the technical aspects of various railway network components, such as *bridges, tunnels*, and *railroads*, and the use of *computer aided design* to facilitate/support *decision making* on the *project management* side, including *cost engineering*, *quality control*, *lean construction*, and *innovation*.

Moreover, the results indicated that from 2017 onwards, there was an increased emphasis on exploring the theory and application/integration of emerging technologies in *railway construction* and *asset management*. The theoretical research focused on *information theory, fuzzy logic, artificial intelligence, interoperability,* and *systems integration*. On the other

hand, the application/integration research delved into the advancements of *BIM technology*, such as *5D-BIM* and *digital twins*, as well as *big data* and *geographic information systems (GIS).*

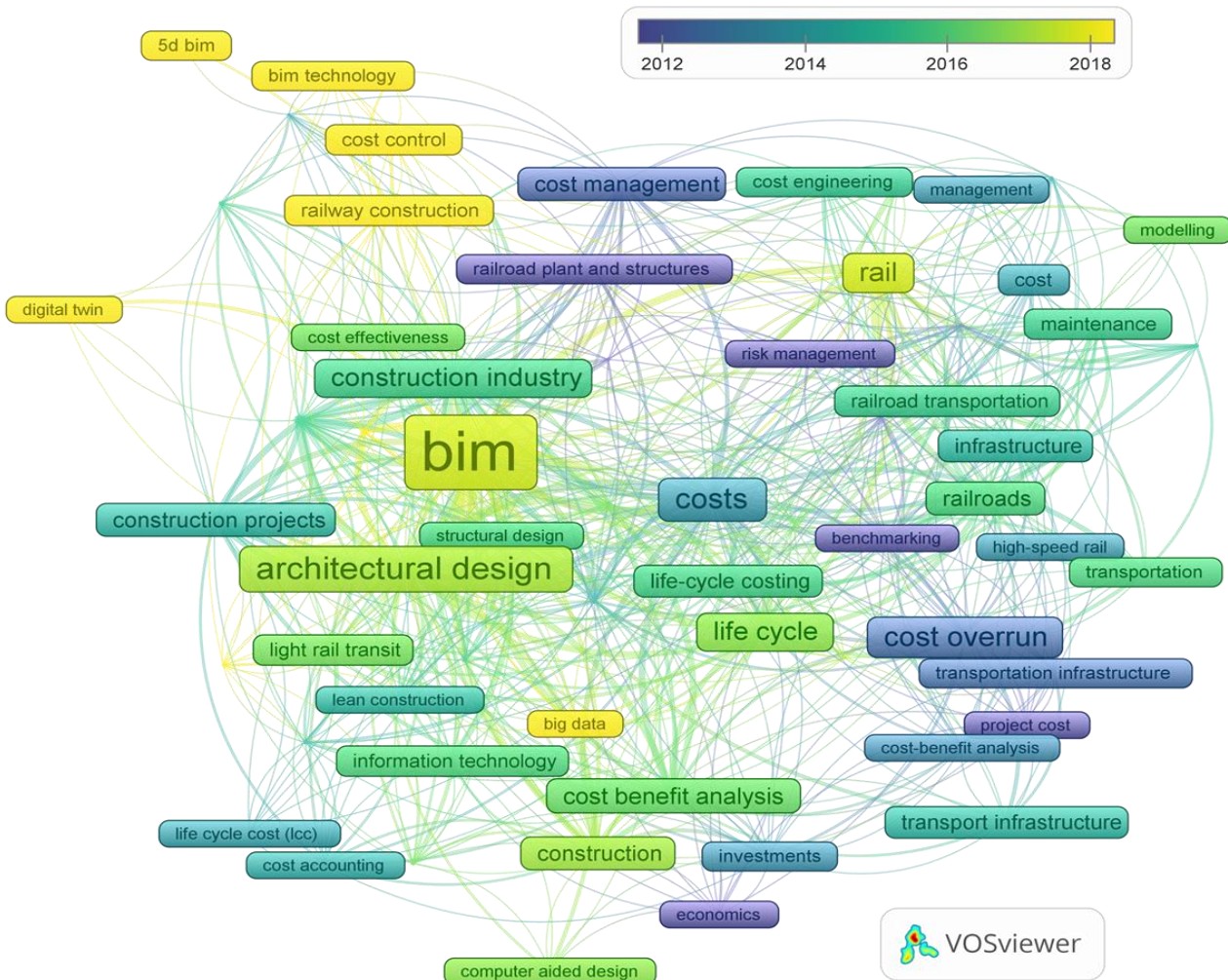

**Figure 14.** Time trend overlay visualization based on 380 records.

## 6. Conclusions

In conclusion, this systematic literature review has provided insights into the current state of 5D-BIM implementation in rail projects, identified research gaps and trends, and highlighted the potential of 5D-BIM to address cost overrun issues. The study analyzed various aspects of cost overrun, including causes, estimation methods, management and control strategies, and the applications of 5D-BIM in rail projects. The study also identified technical, personal, and process challenges associated with 5D-BIM implementation.

Based on the 1888 papers, the study presented, analysed and discussed the trend of BIM publications in the last 23 years, key journals, influential authors, and top contributing nations in the research field.

The analysis has revealed eight clusters: railway infrastructure, BIM, construction projects, demand forecast, cost-effectiveness analysis, supply chain, cost deviation, construction contract, cost control, and troubled projects. These clusters were carefully aggregated and analysed to address the research questions. The key findings against the research questions could be summarized as follows:

1.  The study underscored the considerable conceptual controversy regarding the directionality of cost overrun, its definitions, and the diversity of perspectives and underlying theories. The study found that cost overrun causes could be dependent

on the viewpoint, with auditors explaining cost overruns as technical challenges, the economic, psychological and political literature focusing on the perspective of the public decision maker, and construction engineering managerial analysis focusing on contractual incompetence and related technical consequences.

2.  The study also analysed various quantitative and qualitative cost estimation methods employed in transportation and rail projects, with the rail industry primarily relying on parametric, artificial neural network (ANN), and Monte Carlo simulation-based techniques. Qualitative approaches used in rail projects, such as the analytical hierarchy process (AHP), artificial neural network (ANN), fuzzy neural network (NN), case-based reasoning (CBR), and expert judgment (EJ), depend on the estimator's understanding of the project and the scope of work, while quantitative methods, such as unit cost, analytical hierarchy process (AHP), BIM, graphical evaluation and review technique (GERT), program evaluation and review technique (PERT), structural equation modeling (SEM), and regression analysis (RA), rely on historical data collection and analysis.

3.  The study further revealed that despite extensive research efforts and the implementation of various cost management and control strategies, such as reference class forecasting (RCF), data mining, historical data analysis, and contingency planning, most of these strategies have significant limitations and theoretical flaws. Therefore, the study emphasizes the potential of recent advancements in 5D-BIM to address the root causes of the problem.

4.  The study also examined the various applications of 5D-BIM in rail and transport projects, identifying its use in quantity take-off, cost estimation, cost budgeting, cost control, and lifecycle cost analysis. The benefits of BIM at different stages of the typical rail project lifecycle were identified, including creating a unified platform for data storage and management during the survey stage, design visualization and collaborative work during the design stage, schedule and site management during the construction stage, and operation and disaster emergency simulation in the operation stage. Alongside the BIM benefits, the study identified technical, personal, and process challenges for BIM implementation in rail projects.

The results of this study can be valuable for both researchers and practitioners in the field of rail project management. For researchers, this study provides a comprehensive overview of the current state of the 5D-BIM research field and highlights the most relevant topics for future research. Practitioners can benefit from the insights into 5D-BIM implementation in practical settings, including the benefits and challenges associated with its use.

To fully harness the capabilities of 5D-BIM implementation, a robust framework that considers BIM policies and standards, tools and techniques, and overall project governance is necessary. Additionally, prevailing cost estimation and management techniques, which are driven by professional standards in the rail industry, should also be taken into consideration.

Finally, while limitations associated with the specific keywords and databases chosen for this systematic literature review exist, the main themes and trends identified are expected to remain relevant. Overall, this study provides valuable insights for researchers and practitioners looking to deploy 5D-BIM to minimize cost overruns in rail projects.

**Author Contributions:** Conceptualization and methodology, all authors; software and visualization O.A.I.H.; project administration, writing—original draft preparation, funding acquisition, investigation, validation, resources, data curation, and formal analysis, O.A.I.H. and R.C.M.; writing—review and editing, all authors; supervision, R.C.M., S.D.C.W. and D.D.A.-D. All authors have read and agreed to the published version of the manuscript.

**Funding:** This research and APC was funded by Monash University Graduate Scholarship.

**Institutional Review Board Statement:** Not applicable.

**Informed Consent Statement:** Not applicable.

**Data Availability Statement:** The data employed in this review were downloaded from Scopus, Science Direct, Web of Science, and Google Scholar. The data could be replicated using the search strategy in Appendix G.

**Acknowledgments:** The authors would like to acknowledge the financial assistance provided by Monash University, Clayton, Australia under the Monash Graduate Scholarship.

**Conflicts of Interest:** The authors declare no conflict of interest.

## Abbreviations

The below abbreviations are used in this paper:

| | |
|---|---|
| 5D-BIM | Five-Dimensional Building Information Modelling |
| AACEI | Association for the Advancement of Cost Engineering International |
| ABC | Activity-Based Costing |
| AECO | Architecture, Engineering, Construction, and Operations |
| AHP | Analytical Hierarchy Process |
| AI | Artificial Intelligence |
| ANNs | Artefactual Neural Networks |
| BIM | Building Information Modeling |
| CBR | Case-Based Reasoning |
| CBS | Cost Breakdown Structure |
| EVM | Earned Value Management |
| EJ | Expert Judgment |
| FIG | International Federation of Surveyors |
| GA | Genetic Algorithm |
| GIS | Geographic Information System |
| GERT | Graphical Evaluation and Review Technique |
| ICEC | Cost Engineering Council |
| ICMS | International Cost Management Standard |
| IFC | Industry Foundation Classes |
| IoT | Internet of Things |
| KPIs | Key Performance Indicators |
| LCC | Life-Cycle Costs |
| LCCA | Life-Cycle Cost Analysis |
| ML | Machine Learning |
| MRA | Multiple Regression Analysis |
| NRM | New Rules of Measurement |
| PBO | Parliamentary Budget Office |
| PRISMA | Systematic Reviews and Meta-Analyses |
| RBC | Resource-Based Costing |
| RCF | Reference Class Forecasting |
| RA | Regression Analysis |
| RICS | Royal Institution of Chartered Surveyors |
| SLR | Systematic Literature Review |
| SEM | Structural Equation Modeling |
| TCM | Total Cost Management |
| TVD | Target Value Design |
| WBS | Work Breakdown Structure |

## Appendix A. Different 5D-BIM Uses as Discussed in the Literature

| Research Method | Source | Region | Project Phase | Perspective | Purpose of Using 5D-BIM | | | | Industry | | | Conclusion/Findings |
|---|---|---|---|---|---|---|---|---|---|---|---|---|
| | | | | | Quantity Take-Off (Quantification) | Cost Estimation | Cost Monitoring and Control | Lifecycle Cost Analysis | Rail/Transport | Infrastructure | Construction | |
| Case study | Digital project management in infrastructure project: a case study of Nagpur Metro Rail Project [21] | India | Construction | Contractor | X | X | X | X | X | - | - | The deployment of a BIM-based integrated digital project management system in the Nagpur Metro Rail Project has benefited the project in a variety of ways, including improved cost management and control. |
| Case Study | 5D-BIM applied to cost estimating, scheduling, and project control in underground projects [298] | Europe | Construction | Client | X | - | X | - | X | - | - | 5D-BIM is highly recommended in case of alternative project delivery such as design–build and P3 Projects. BIM's best added value is appreciated in complex projects such as urban tunnelling and complex projects such as underground hydropower plants, railway and highway twin tube tunnel projects and repository underground structures. |
| Questionnaire | Benefits of integrating 5D-BIM in cost management practices in quantity surveying firms [299] | Nigeria | ALL | - | X | X | X | X | - | - | X | Cost managers will benefit from 5D-BIM in a variety of ways, including automated quantity take-off and improved project visualisation during the design and construction stages. |
| Case Study | Time and cost control of construction project using 5D- BIM process [300] | India | Construction | Client | X | X | X | | - | - | X | 5D-BIM provides various advantages in terms of time and cost management for building projects, including faster procurement process, precise/fast decision making. |
| Review | Analysis on the BIM application in the whole life cycle of railway engineering [218] | China | ALL | Client | - | - | - | X | X | - | - | BIM technology will progressively advance railway construction and, in the near future, will replace CAD. It will propel railway construction to a greater degree of informatization and intelligence growth. |
| Modelling simulation of a railway station | Digital twin for sustainability evaluation of railway station buildings [188] | UK | Construction | Client | X | X | X | X | X | - | - | The adoption of BIM in railway station construction projects provides several benefits. |
| Review | Overview: the opportunity of BIM in railway [22] | Morocco | ALL | - | X | X | X | X | X | - | - | BIM integration in rail is becoming a worldwide trend. This integration requires government decisions, more political impulse and a maturation of technology and tools. |
| Case study | Applying building information modelling to integrate schedule and cost for establishing construction progress curves [301] | Taiwan | Construction | Client | X | X | X | X | - | - | X | A four-step model incorporating BIM objects was used to establish a construction S-Curve. |
| Case study | Research on cost control of construction project based on the theory of lean construction and BIM: Case Study [302] | China | Construction | Client | - | - | X | - | - | - | X | Demonstrated by a case study, it is shown how a combination of lean theory and BIM can improve cost control in construction projects. |

| Research Method | Source | Region | Project Phase | Perspective | Purpose of Using 5D-BIM | | | | Industry | | | Conclusion/Findings |
|---|---|---|---|---|---|---|---|---|---|---|---|---|
| | | | | | Quantity Take-Off (Quantification) | Cost Estimation | Cost Monitoring and Control | Lifecycle Cost Analysis | Rail/Transport | Infrastructure | Construction | |
| Case study | Implementing earned value management using bridge information modelling [303] | Egypt | Construction | Client | X | X | X | X | - | - | X | Presented a case study for the application of BIM in cost and time management of infrastructure bridge. |
| Case study | Project cost control using five dimensions building information modelling [304] | Egypt | Construction | Contractor | X | X | X | - | - | - | X | Using 5D-BIM improves project financial decision making (including stakeholder communications, cost estimation and control process). |

## Appendix B. Cost Estimation Methods (Models)

| Approach | Category | Cost Estimation Method (Model) | Description |
|---|---|---|---|
| Quantitative | Parametric | Regression Analysis (RA) | Regression analysis is a statistical technique used to investigate the relationship between variables [305]. It provide simple analysis to sort out the impact of different parameters on the project costs [306]. |
| | | Monte Carlo Simulation (MCS) | Monte Carlo simulation uses random sampling and statistical modelling to estimate mathematical functions and simulate the processes of complex systems [307]. Monte Carlo simulation is used to calculate contingency and cost estimate uncertainties [308]. |
| | | Structural Equation Modelling (SEM) | SEM is a comprehensive statistical method that tests hypotheses about relations between observed and latent variables [309]. SEM is a combination of two statistical methods: confirmatory factor analysis and path analysis [310]. |
| | | Program Evaluation and Review Technique (PERT) | PERT uses random variables with the following parameters to estimate the cost/duration of an activity: a—optimistic cost/time required to accomplish a task, m—the most probable cost/time required to accomplish a task, b—pessimistic cost/time required to accomplish a task. The value of estimated cost/time is equal to ((a + 4m + b)/6)) [311] |
| | | Graphical Evaluation and Review Technique (GERT) | GERT was introduced by [312]. It is a technique used to analyse stochastic networks that contain activities with a probability of occurrence associated with them, and treat the plausibility that time/cost required to complete an activity is a random variable (not a constant) [313]. |
| | Analytical | Decision Tree | Decision tree approach is a popular data mining method for constructing prediction algorithms for a target variable or establishing classification systems based on many variables [314]. |
| | | BIM | BIM object-oriented system helps facilitate generating bottom-up estimates and quantity take-off [315]. |
| | | Unit Cost | The unit cost estimate method focuses on determining the cost of materials, equipment, and labour for each component of a construction, which requires a detailed quantities take-off [316]. |
| Qualitative | Intuitive | Analytical Hierarchy Process (AHP) | AHP is a decision-making support approach for selecting a solution from alternatives based on a set of evaluation criteria [317]. |
| | Analogous | Artificial Neural Network (ANN) | A neural network simulates the operation of the human brain. It excels at tackling complicated non-linear mathematical problems [318]. |
| | | Fuzzy Neural Network (NN) | Neural networks (NNs) are modelled after biological neural systems [319], while fuzzy logic is a tool for simulating human cognition and perception [319]. It describes process uncertainties and imprecision [320]. Combined together can form powerful tool to estimate project costs [320]. |
| | | Case-Based Reasoning (CBR) | CBR is an approach for solving new problem cases by reusing findings from old cases. The CBR systems consist of a data base to store old cases along with their solutions [321]. |
| | | Expert Judgment (EJ) | Expert judgement (EJ) approach relies on the understanding/thinking and reasoning of experts on processing historical cost data to make sound judgment on project cost [322]. |
| | | Support Vector Machine (SVM) | SVM is a computer algorithm that uses examples to learn how to label objects [323]. SVM can be used in different ways to support the estimation process [324,325] |

### Appendix C. Key Publications Covering Cost Overrun Causes in Transport and Rail Projects

| Research Method | Source | Region | Project Phase | Industry | | Cost Overrun | | Conclusion/Findings |
|---|---|---|---|---|---|---|---|---|
| | | | | Rail | Transport | Perspective | Category | |
| A study based on a sample of 258 transportation infrastructure projects. | Underestimating costs in public works projects: Error or lie? [40]. | USA | ALL | - | X | Client | Psychological/Technical | Cost underestimation cannot be explained by error and seems to be best explained by strategic misrepresentation, i.e., lying. In 9 out of 10 transportation infrastructure projects, costs are underestimated. For rail projects, actual costs are on average 45% higher than estimated costs. Cost underestimation exists across 20 nations and 5 continents; it appears to be a global phenomenon. |
| Investigated the causes of project cost overruns reported in the construction-management-related articles since 1985. | Review of construction journals on causes of project cost overruns [326]. | Worldwide | ALL | X | X | - | Technical/Economic/ Psychological/Political | The study identified 79 causes of cost overruns, the top causes that have received the highest number of citations includes: design problems, inaccurate estimation, poor planning, poor communication, and poor financial management. |
| A study based on a sample of 258 transportation infrastructure projects. | What causes cost overrun in transport infrastructure projects? [142]. | USA | ALL | X | X | Client | Technical/Economic/ Psychological/Political | Cost escalation is highly dependent on length of project implementation phase. Data do not support that bigger projects have a larger risk of cost escalation than do smaller ones. Public projects are not more problematic compared to privately owned projects (in terms of cost overrun). |
| Case study | Cost overruns and delays in infrastructure projects: the case of Stuttgart 21 [327]. | Germany | All | X | X | Client | Technical/Economic | Cost overrun causes include: scope changes, geological conditions, high risk-taking propensity, extended implementation, price overshoot, conflict of interests and lack of citizens' participation. |
| Case studies | Cost overruns in Australian transport infrastructure projects [273]. | Australia | All | X | X | Client | Technical | Studied the magnitude of cost overruns on Australian transport infrastructure projects. |
| Literature study | Cost overruns in large-scale transportation infrastructure projects: Explanations and their theoretical embeddedness [143]. | Worldwide | All | - | X | Client | Technical/Economic/ Psychological/Political | Discussed agency theory, eclectic theory, rational choice theory and prospect theory. |
| Statistical analysis of case studies. | Cost overruns in road construction—what are their sizes and determinants? [42]. | Norway | All | - | X | Client | Technical | Investigated the statistical relationship between actual and estimated cost, cost overrun is found to be more predominant as compared to cost savings, there are significant number of projects being completed with actual costs less than estimated. Provided policy implications. |

| Research Method | Source | Region | Project Phase | Industry | | Cost Overrun | | Conclusion/Findings |
|---|---|---|---|---|---|---|---|---|
| | | | | Rail | Transport | Perspective | Category | |
| Analysed government project data. | On the magnitude of cost overruns throughout the project life cycle: An assessment for the Italian transport infrastructure projects [272]. | Italy | All | - | X | Client | Technical | Analysed government project data and the whole process of cost generation for transport infrastructure works. |
| Analysed rail project data set. | Cost overrun and demand shortfalls in urban rail and other infrastructure [144]. | Worldwide | All | X | X | Client | Technical/Economic/ Psychological/Political | The analysis of construction costs shows that urban rail projects on average turn out to be substantially more costly than forecast. At the same time, the analysis of ridership shows urban rail to achieve considerably fewer passengers than forecast and thus lower revenues. The article showed that urban rail projects are particularly risky ventures, although other transportation projects, such as tunnels and bridges, are also highly risky, as are projects in policy areas other than transportation: Average cost escalation for urban rail is 45% in constant prices. For 25% of urban rail projects cost escalations are at least 60%. Actual ridership is on average 51% lower than forecast. For 25% of urban rail projects, actual ridership is at least 68% lower than forecast. When cost risk and revenue risk are combined, a risk profile emerges for urban rail, which proves such projects to be economically risky to the second degree. |
| Analysed a data set of 1091 transport projects developed by the Portuguese government. | The determinants of cost deviations and overruns in transport projects, an endogenous model's approach [328] | Portugal | All | - | X | Client | Technical | Profound implications concerning public policy, because when undertaking large infrastructure developments plans, and estimating their potential cost (and overruns), it is fundamental to understand the current economic dynamics, as well as acting on improving the overall legal (particularly regarding public procurement laws) and governance environment, particularly regarding the government's efficiency, corruption, and the overall rule of law. |
| Investigated the risk factors leading to substantial cost overruns of highway projects and develop a more definitive risk contingency allocation regime for overall highway projects to supersede the arbitrary models currently present. | Evaluation of risk factors leading to cost overrun in delivery of highway construction projects [265] | Australia | project development. | - | X | Client | Technical | Investigated the statistical models that can explain the correlation between the cause, effect, and other relationships relating to the cost overrun in highway construction projects. The regression analysis demonstrated a weak correlation between the size of highway projects, as measured in the indexed programmed cost, and the size of cost overruns. It can also be concluded from the research that the arbitrary application of a base contingency percentage figure, such as 10%, to accommodate project risk can lead to those projects reporting a substantial budget overrun. |
| Analysed a project data set (a sample of 258 projects worth approximately USD 90 billion). | How common and how large are cost overruns in transport infrastructure projects? [145] | USA | ALL | X | X | Client | Technical/Economic/ Psychological/Political | Cost estimates used in public debates, media coverage and decision-making for transport infrastructure development are highly, systematically, and significantly deceptive. The risks generated from misleading cost estimates are typically ignored or underplayed in infrastructure decision-making. |

| Research Method | Source | Region | Project Phase | Industry | | Cost Overrun | | Conclusion/Findings |
|---|---|---|---|---|---|---|---|---|
| | | | | Rail | Transport | Perspective | Category | |
| Analysed a project data set (a sample of 78 projects). | Characteristics of cost overruns for Dutch transport infrastructure projects and the importance of the decision to build and project phases [216]. | Netherlands | ALL | - | X | Client | Psychological/Political | Found that cost overruns have been a problem for the last 20 years. Furthermore, although in the Netherlands cost overruns are about as common as cost underruns, the average overrun is larger than the average underrun. Overall, projects have an average overrun of 16.5%. Considering these findings, rejecting technical explanations, the cost underestimation in Dutch projects can better be explained by psychological and political–economic explanations. The most common psychological explanation is probably "appraisal optimism". |
| Literature study | How to Build Major Transport Infrastructure Projects within Budget, in Time and with the Expected Output; a Literature Review [286]. | Worldwide | ALL | - | X | Client | Technical/Economic/ Psychological/Political | The main conclusion from the review is that in the current scientific literature on major transportation infrastructure projects, four main factors are mentioned that might help to build these projects in time, on budget and with the expected output: improving cost and benefit estimates, risk-containment measures, increasing accountability, and clear scope and objectives. |
| Analysed a project data set (a sample of 78 projects) | Different cost performance: Different determinants? The case of cost overruns in Dutch transport infrastructure projects [329]. | Netherlands | ALL | X | X | Client | Technical | The study showed that in the Netherlands, cost overruns for rail projects are relatively low, both when compared nationally with roads and fixed links, and internationally when compared with worldwide findings. The difference between project types may be related to the organisational set-up and institutional settings, which is different for rail projects (with ProRail as project owner) and for road projects (with RWS as project owner). This research furthermore concluded that small projects have the largest average cost overrun. This suggests that smaller projects deserve more attention than is currently the case, as they result in similar percentage cost overruns as the large projects. |
| Systematic Literature Review | Tales on the dark side of the transport infrastructure provision: a systematic literature review of the determinants of cost overruns [5]. | Worldwide | | - | X | Different perspectives | Technical/Economic/ Psychological/Political | This study provides a systematic review of the broad and heterogeneous literature that investigates the determinants of cost overruns in transport infrastructure provision. It focuses on empirical analyses, published between 2000 and 2016. |
| Case studies | Cost overruns in Swedish transport projects [330]. | Sweden | ALL | X | X | Client | - | A good strategy to improve cost calculation would be to develop a cost estimation method which considers the risks of the costs in each individual component based on the experiences of a class of similar projects. This is the same concept as the risk-based estimating method used in Australia. It combines advantages from both the successive calculation and the reference class forecasting method. |
| Literature study | Debunking fake news in a post-truth era: The plausible untruths of cost underestimation in transport infrastructure projects [9]. | Worldwide | ALL | - | X | Client | - | A detailed examination of the Flyvbjerg, Holm and Buhl research raises serious questions regarding the methodology adopted, the analysis undertaken, and unfounded conclusions reached. |

| Research Method | Source | Region | Project Phase | Industry | | Cost Overrun | | Conclusion/Findings |
|---|---|---|---|---|---|---|---|---|
| | | | | Rail | Transport | Perspective | Category | |
| Critical analysis | Explaining cost overruns of large-scale transportation infrastructure projects using a signalling game [331]. | Worldwide | Biding | - | X | Client/Contractor | Political-economic | The signalling game gives useful insights into the way in which strategic behaviour results in cost underestimation. It is, furthermore, a valuable tool to predict the impact of policy measures on the behaviour of the market party. Measurements are aimed to reprimand or prevent the strategic behaviour of the market party and they should be focused on changing the incentive structure in such a way that the signal of the game becomes effective. |
| Critical analysis/ Literature study | Cost overruns in transportation infrastructure projects: Sowing the seeds for a probabilistic theory of causation [105]. | Worldwide | All | - | X | Client | | Probabilistic causal inferences about cost overruns can be acquired from a combination of assumptions, experiments, and data. |
| Review | Toward a Systemic View to Cost Overrun Causation in Infrastructure Projects: A Review and Implications for Research [6]. | Worldwide | All | - | X | - | - | Explored some of the methodological deficiencies in the approaches adopted in a majority of the cost overrun research. These deficiencies include a poor understanding of systemicity and embeddedness of the sources of overruns, a dependence on correlational analysis, a lack of demonstrable causality, and superficiality of the research design. Found that cost overrun research has largely stagnated in the refinement and advancement of the knowledge area; the bulk of it has largely been replicative. |
| Critical analysis/ Literature study | On de-bunking "fake news" in a post truth era: Why does the Planning Fallacy explanation for cost overruns fall short? [146]. | Worldwide | All | - | X | Client | - | Critically questioned the work presented by Bent Flyvbjerg. |
| Analysed a project data set | Cost Overrun and Cause in Korean Social Overhead Capital Projects: Roads, Rails, Airports, and Ports [116]. | Korea | All | - | X | Client | - | In Korea, the causes of cost overruns can be grouped into several major categories: changes in the scope of a project, delays in construction, unreasonable estimations and adjustments of the project costs, and no practical use of the earned value management system. |
| Critical literature review | Construction Projects Cost Overrun: What Does the Literature Tell Us? [242]. | Worldwide | All | - | X | - | - | 173 causes of cost overrun have been found in seventeen contexts, with the main potential causes being: frequent design change, contractors' financing, payment delay for completed work, lack of contractor experience, poor cost estimation, poor tendering documentation, and poor material management. |
| Systematic literature review | Cost Overrun Causative Factors in Road Infrastructure Projects: A Frequency and Importance Analysis [332]. | Worldwide | All | - | X | - | - | It is recommended that the mitigation of cost overruns in road projects be undertaken from the early stages. This due to the fact that several causal factors with high influence values are observed among the top 20 factors with the greatest influence, which are related to different processes that belong to the initial stages of the projects, factors that are under the control of the project stakeholders and therefore have high viability to be addressed. |

| Research Method | Source | Region | Project Phase | Industry | | Cost Overrun | | Conclusion/Findings |
|---|---|---|---|---|---|---|---|---|
| | | | | Rail | Transport | Perspective | Category | |
| Systematic literature review | Systematic Review of Cost Overrun Research in the Developed and Developing Countries [333]. | Developing Countries | All | - | X | - | - | The findings of this study have shown that there have been broad studies conducted on cost overrun in both developing nations and developed nations. However, there is a slight lack in comprehensiveness of cost overrun studies in the developing nations; perhaps future studies on cost overrun in developing nations can be directed to more specific areas of construction projects such as those that have been performed by researchers of the developed nations. |
| Literature review | Academics and Auditors Comparing Perspectives on Transportation Project Cost Overruns [271]. | Worldwide | All | - | X | - | - | There are divergences between the technical and managerial explanations prioritized by the auditors and the political, economic, and psychological explanations prioritized in much of the academic literature. Moreover, the independent government audits place considerably less weight on willful deception and strategic misrepresentation as systematic causes of cost overruns than some of the highest-profile academic studies on the topic [334–336]). These variations are significant, as they point to diverse strategies to reduce the prevalence of cost overruns on future transportation investment projects. |
| Analysed a project data set (Seven large bridge and tunnel projects) | Inaccuracy of traffic forecasts and cost estimates on large transport projects [147]. | Denmark | All | X | X | Client | Technical | Forecasts of project viability for large transport infrastructure projects are often over-optimistic to a degree where such forecasts correspond poorly with actual development. |
| Analysed a project data set (six major European railway projects) | A New Paradigm for the Assessment of High-Speed Rail Projects and How to Contain Cost Overruns: Lessons from the EVA-TREN Project [337]. | Europe | All | X | X | - | - | Highlighted cost overrun and lessons learned from EVA-TREN Project. |
| Analysed a project data set (Sixteen rail projects) | Trends in U.S. rail transit project cost overrun [180]. | USA | All | X | X | Client | - | There is evidence to suggest that cost overruns for projects completed before 1990 are different from that of projects completed after 1994 (i.e., cost overruns have become smaller—positive trend). |

**Appendix D. Cost Models Used for Cost Estimation, Prediction, and Analysis in Transport and Rail Projects**

| Cost Estimation Method/Model | Source | Region | Project Phase | Industry | | Conclusion/Findings |
|---|---|---|---|---|---|---|
| | | | | Rail | Transport | |
| Earned value | The control model of engineering cost in construction phase of high-speed railway [338] | China | Construction | X | X | To improve the efficiency of cost control in the high speed railway construction phase, the researchers set up the model of earned value and install FBCWS index, through the contrast between FBCWS index and ACWP index, they have improved the efficiency of cost control in construction stage, so that they can do the better in the direction and control before costs incurred and make the construction cost control management more scientific and effective in the construction phase of high speed rail project. |
| Life Cycle Costing | An application of a generalized life cycle cost model to boxn wagons of Indian railways [339] | India | Operation and Maintenance | X | X | A generalized life cycle cost model for repairable and non-repairable products based on reliability and maintainability (M) aspects is applied to BOXN wagons used by Indian railways and the results obtained are discussed. |
| Multiple | Cost Estimation Methods for Transport Infrastructure: A Systematic Literature Review [284] | Worldwide | All | X | X | According to the SLR, 12 different cost estimation methods have been used in different transport infrastructure modes. Among these, the parametric method has been used the most, followed by artificial neural networks. With respect to infrastructure type, the focus was mostly on roads. The trend shows that research on cost estimation methods has been increasing over the years and more types of methods are being used. Most of the research found focused on the experimental use of different methods, and not the analysis of the methods practiced in the industry. |
| Case-based Reasoning (CBR) estimate | The Approximate Cost Estimating Model for Railway Bridge Project in the Planning Phase Using CBR Method [282] | Korea | Planning | X | X | Suggested the cost estimation model which uses CBR and makes the database reflect the character of the railroad bridge. The study examined combinations of attributes, criteria of similarities, and retrieval ranks and applied GA for an optimization of attribute weights throughout learning process. |
| Linear Regression Analysis and Artificial Neural Networks (ANNs) | Cost and Material Quantities Prediction Models for theConstruction of Underground Metro Stations. [340] | Greece | Construction | X | X | Using linear regression analysis and ANNs in comparing the actual values of costs and quantities with the corresponding predictions proved to be efficient and reliable cost estimation methodology. |
| Multiple regression analysis | Early cost estimation models based on multiple regression analysis for road and railway tunnel projects [341] | Western Europe | Planning | X | X | Developed tunnel cost estimation models that can be used for various applications in the planning stage of road and railway projects. The models were developed using data from 25 constructed projects in western European countries. |
| BIM | Optimization of cost of a tram through the integration of BIM: A theoretical analysis [260] | Morocco | Construction | X | X | Conducted a theoretical analysis of the optimization of the cost of a tram by integrating the building information modelling (BIM) from the sketching phase and throughout the life cycle of the infrastructure. The analysis showed that BIM would reduce 8.4% of the overall cost of a tramway project. It also showed that BIM would save 10% of maintenance costs over 30 years. |
| Life Cycle Costing | Development of a life cycle cost estimate system for structures of light rail transit infrastructure [211] | Korea | Construction | X | X | An LRT-LCC system was developed in this study, based on existing studies on LRT construction cost estimation and LCC estimation studies for bridges, tunnels, and buildings. The system was composed to provide a feasibility analysis based on the existing economic analytical results of each structure required for LRT construction. |
| Pairwise comparisons | Modelling the cost of railway asset renewal projects using pairwise comparisons [342] | UK | Design | X | X | Presented the development process of a cost-estimating model for railway renewal projects at the early stage of a project life cycle. The practical implications of the developed model are its ability to estimate renewal project costs of railway assets when there is a lack of quantitative data and detailed project definition. |

| Cost Estimation Method/Model | Source | Region | Project Phase | Industry | | Conclusion/Findings |
|---|---|---|---|---|---|---|
| | | | | **Rail** | **Transport** | |
| Statistical methods | Determining the Probability of Project Cost Overruns [343] | Australia | All | - | X | Developed a Fréchet probability function that can be used to calculate the probability of cost overruns. |
| Parametric cost estimation | Parametric cost estimation system for light rail transit and metro track works [344] | Turkey | Concept | X | X | Developed a multivariable regression and artificial neural network models for cost estimation of the construction costs of track works for light rail transit and metro projects at the early stages of the construction process. |
| Present Worth Analysis, Internal Rate of Return and Cost-Benefit Analysis | Railway Investment Appraisal Techniques [345] | Europe | Concept | X | X | Presented the basic principles and applications of the most important investment appraisal techniques in a clearly written fashion, supported by a number of railway-related examples. |
| A set of cost functions | A tool for railway transport cost evaluation [346] | Italy | Feasibility study | X | X | Provided a systematic process for cost estimation and decision support. The methodology can be used as an intermediate tool to allow rail planners to more easily perform railroad analysis and planning activities on their own, prior to contracting out feasibility studies. |

### Appendix E. Popular Methods/Techniques Used for Cost Overrun Prediction, Cost Estimate and Cost Contingency Calculations

| Method/Technique | Type | Definition | Method Uses from the Literature | | |
|---|---|---|---|---|---|
| | | | Cost Overrun Prediction | Cost Contingency Calculations | Cost Estimation |
| **Case-based reasoning (CBR)** | Analogical method | "A case-based reasoner solves new problems by adapting solutions that were used to solve old problems." [347] | [276] | | [348,349] |
| **Multiple regression analysis (MRA)** | Statistical method | "Multiple regression is used as a data-analytic strategy to explain or predict a criterion (dependent) variable with a set of predictor (independent) variables" [350] | [351] | [352–354] | [355,356] |
| **Artificial neural networks (ANN)** | Repetitive learning | "A massively parallel combination of simple processing unit which can acquire knowledge from environment through a learning process and store the knowledge in its connections." [357] | [351] | | [353,354] |
| **Monte-Carlo simulation (MCS)** | Stochastic method | "The Monte Carlo method is an application of the laws of probability and statistics to the natural sciences" [358] | | | [359] |

### Appendix F. Systematic Literature Review (SLR) Protocol

Minimizing Cost Overrun in Rail Projects Through 5D-BIM: A Systematic Literature Review.

Review Protocol.

| | |
|---|---|
| Organization, city, country | Monash University, Melbourne, Australia |
| Prepared by | Osama Hussain |
| Date | Updated on 8 January 2023 |
| Review team members | Dr. Robert Moehler—Monash University Dr. Stuart Walsh—Monash University |

*Appendix F.1. Background*

This systematic literature is the first step of a broader research to investigate the use of 5D-BIM modelling to minimize cost overrun in rail projects.

The research will consider different cost management and control frameworks that employ 5D-BIM and evaluate their impact on cost control. It aims to produce a framework and guide on the best practices for using BIM to control cost overruns in the rail industry, with a long-term goal of informing regulation and policy.

*Appendix F.2. Objective*

The objective of this systematic literature review (SLR) is to give a quick, detailed overview of the literature and the main trends. The SLR will be used to map the knowledge gaps and synthesise the existing body of knowledge [360–362].

*Appendix F.3. Researchers*

First reviewer: Osama Hussain—Monash University—Department of Civil Engineering
Osama.hussain@monash.edu
Review Team members:

- Dr. Robert Moehler—Monash University

  Robert.Moehler@monash.edu

- Dr. Stuart Walsh—Monash University

  Stuart.Walsh@monash.edu

*Appendix F.4. Research Questions*

The specific review questions to be addressed are as follows:

1. What causes cost overrun in transport projects in general and rail projects in particular?
2. What are the cost models used to predict and analyse cost overrun in transport projects in general and rail projects in particular?
3. What cost management and control strategies are used to prevent these cost overruns? What is the efficiency of these strategies and suitability for 5D-BIM modelling?
4. How can 5D-BIM be successfully integrated into rail projects life cycle to support cost management and control models and minimize/prevent cost overrun?
5. What is the validity and reliability of using 5D-BIM modelling for different types of rail projects?

*Appendix F.5. Time Line for the Review*

| No | Stage | Duration |
|---|---|---|
| 1 | Protocol | 2$^1$/$_2$ weeks |
| 2 | Literature searching | 2 weeks |
| 3 | Screening/Quality appraisal | 2 weeks |
| 4 | Data extraction | 6 weeks |
| 5 | Synthesis | 4 weeks |
| 6 | Writing up | 4$^1$/$_2$ week |
| | Total | 21weeks |

*Appendix F.6. Electronic Databases*

The following electronic databases will be used for data collection.

| **Main Sources** | | |
|---|---|---|
| 1 | Scopus | https://www.scopus.com (accessed on 8 January 2023) |
| 2 | Science Direct | https://www.sciencedirect.com (accessed on 8 January 2023) |
| 3 | Web of Science (new website) | https://www.webofscience.com/wos/woscc/basic-search (accessed on 8 January 2023) |
| **Additional sources** | | |
| 4 | Google Scholar | https://scholar.google.com (accessed on 8 January 2023) |

Note: The search algorithm for Google scholar is not known and cannot be controlled. Google adapts the search to each user in order to personalize information and, as a result, a systematic search is quite probably not replicable [128]. Google Scholar was considered as an additional source only for this systematic literature review.

*Appendix F.7. Inclusion/Exclusion Criteria*

| Area | Inclusion | Exclusion |
|---|---|---|
| Databases | Indexed in: Scopus, Science Direct, Web of Science and Google Scholar | Not indexed in: Scopus, Science Direct, Web of Science and Google Scholar |
| Document Types | Journal articles, conference papers, books, and theses. | All other types of publications |
| Years | 2000–2023 | Prior to 2000 |
| Language | English | Non-English |

For detailed search inclusions/exclusions, please refer to search strategy.

*Appendix F.8. Search Strategy*

The search strategy will be designed to access published material in the electronic databases as follows:

(Please refer to the search strategy for details).

1.  Search in Scopus, Science Direct, and the new website for Web of Science will be conducted using keyword to identify cluster and specific words to identify research focus (in all fields).
2.  Search in Google Scholar will be conducted using specific words in the article title.

*Appendix F.9. Tools and Software Packages*

Preferred reporting items for systematic reviews and meta-analyses (PRISMA), guidelines and flowchart will be used for this systematic review [121].

A combination of software packages will be used for data collection, processing and exporting. These include: EndNote [125], Mendeley [126], Excel [127], VOSviewer [122], and Covidence [123].

*Appendix F.10. Screening /Quality Appraisal*

Identified articles that meet the criteria will be grouped into one of the following categories:

Cost overrun, Cost management and control, BIM, and Rail projects. These articles will then be assessed independently by two reviewers.

A clearer definition of inclusion and exclusion criteria will be written based on discussion and agreements. Any disagreements that arise between the reviewers will be resolved through discussion and with the assistance of a third reviewer where required. Screening steps will be as follows:

1.  Title/abstract review: Determine relevancy to the subject area.
2.  Full text review: Verification of the decision of inclusion performed in the first step.

*Appendix F.11. Data Extraction*

Data will be extracted and exported in different formats for further processing. VOS Viewer will be used to visualize the data.

**Appendix G. Search Strategy**

(8 January 2023)

**Search query string**

Scopus search

(1—Cost Overrun)

(KEY ("cost overrun" OR "cost overruns" OR "cost escalation" OR "budget overrun" OR "cost growth" OR "cost underestimation") AND TITLE-ABS-KEY ("transport" OR "rail" OR "railway" OR "BIM" OR "5D BIM" OR "cost management" OR "cost control" OR "project" OR "cost model" OR "causes" OR "sources" OR "driver" OR "life cycle cost")) AND DOCTYPE (ar OR cp) AND PUBYEAR > 1999 AND PUBYEAR > 2000 AND PUBYEAR < 2024 AND (EXCLUDE (SUBJAREA, "ENVI") OR EXCLUDE (SUBJAREA, "EART") OR EXCLUDE (SUBJAREA, "MATH") OR EXCLUDE (SUBJAREA, "MATE") OR EXCLUDE (SUBJAREA, "AGRI") OR EXCLUDE (SUBJAREA, "CHEM") OR EXCLUDE (SUBJAREA, "PHYS") OR EXCLUDE (SUBJAREA, "BIOC") OR EXCLUDE (SUBJAREA, "PSYC") OR EXCLUDE (SUBJAREA, "MEDI")) AND (LIMIT-TO (LANGUAGE, "English"))

(2—Cost Management and Control)

(KEY ("cost management" OR "cost control" OR "project cost management" OR "cost growth" OR "cost underestimation") AND TITLE-ABS-KEY ("transport" OR "rail" OR "railway" OR "BIM" OR "5d BIM" OR "cost model" OR "life cycle cost" OR "strategies" OR " polices" OR "cost overrun" OR "cost overruns" OR "cost escalation" OR "budget overrun")) AND DOCTYPE (ar OR cp) AND PUBYEAR > 1999 AND DOCTYPE (ar OR cp)

AND PUBYEAR > 1999 AND PUBYEAR > 2000 AND PUBYEAR < 2024 AND (EXCLUDE (SUBJAREA, "MEDI") OR EXCLUDE (SUBJAREA, "BIOC") OR EXCLUDE (SUBJAREA, "ENVI") OR EXCLUDE (SUBJAREA, "PHAR") OR EXCLUDE (SUBJAREA, "NURS") OR EXCLUDE (SUBJAREA, "AGRI") OR EXCLUDE (SUBJAREA, "IMMU") OR EXCLUDE (SUBJAREA, "HEAL") OR EXCLUDE (SUBJAREA, "CHEM") OR EXCLUDE (SUBJAREA, "MATH") OR EXCLUDE (SUBJAREA, "MATE") OR EXCLUDE (SUBJAREA, "NEUR") OR EXCLUDE (SUBJAREA, "VETE") OR EXCLUDE (SUBJAREA, "EART") OR EXCLUDE (SUBJAREA, "PSYC") OR EXCLUDE (SUBJAREA, "DENT") OR EXCLUDE (SUBJAREA, "PHYS")) AND (LIMIT-TO (LANGUAGE, "English"))

(3—BIM)

(KEY ("BIM" OR "5d BIM" OR "building information modelling" OR "building information modeling" OR "cost growth" OR "cost underestimation") AND TITLE-ABS-KEY ("transport" OR "rail" OR "railway" OR "cost management" OR "cost control" OR "cost model" OR "life cycle cost" OR "cost overrun" OR "cost overruns" OR "cost escalation" OR "budget overrun")) AND DOCTYPE (ar OR cp) AND PUBYEAR > 1999 AND PUBYEAR > 2000 AND PUBYEAR < 2024 AND (EXCLUDE (SUBJAREA, "BIOC") OR EXCLUDE (SUBJAREA, "MEDI") OR EXCLUDE (SUBJAREA, "ENVI") OR EXCLUDE (SUBJAREA, "EART") OR EXCLUDE (SUBJAREA, "NEUR") OR EXCLUDE (SUBJAREA, "IMMU") OR EXCLUDE (SUBJAREA, "MATH") OR EXCLUDE (SUBJAREA, "MATE") OR EXCLUDE (SUBJAREA, "PHAR") OR EXCLUDE (SUBJAREA, "AGRI") OR EXCLUDE (SUBJAREA, "CHEM") OR EXCLUDE (SUBJAREA, "PHYS")) AND (LIMIT-TO (LANGUAGE, "English"))

(4—Rail projects)

(KEY ("rail" OR "railway" OR "cost growth" OR "cost underestimation") AND TITLE-ABS-KEY ("BIM" OR "5d BIM" OR "cost management" OR "cost control" OR "cost model" OR "life cycle cost" OR "cost overrun" OR "cost overruns" OR "cost escalation" OR "budget overrun")) AND DOCTYPE (ar OR cp) AND PUBYEAR > 1999 AND PUBYEAR > 2000 AND PUBYEAR < 2024 AND (EXCLUDE (SUBJAREA, "ENVI") OR EXCLUDE (SUBJAREA, "MATE") OR EXCLUDE (SUBJAREA, "EART") OR EXCLUDE (SUBJAREA, "PHYS") OR EXCLUDE (SUBJAREA, "MATH") OR EXCLUDE (SUBJAREA, "CHEM") OR EXCLUDE (SUBJAREA, "MEDI") OR EXCLUDE (SUBJAREA, "AGRI") OR EXCLUDE (SUBJAREA, "BIOC") OR EXCLUDE (SUBJAREA, "HEAL") OR EXCLUDE (SUBJAREA, "NEUR")) AND (LIMIT-TO (LANGUAGE, "English"))

Web of science

The new website for (Web of science) was used: https://www.webofscience.com/wos/woscc/basic-search (accessed on 8 January 2023).

(1—Cost Overrun)

**link**

https://www.webofscience.com/wos/woscc/summary/52c989df-d854-4a07-a2e3-bae8240ed200-694a951e/relevance/1 (accessed on 8 January 2023).

Search query string

((((AK=("cost overrun" OR "cost overruns" OR "cost escalation" OR "budget overrun" OR "cost growth" OR "cost underestimation")) AND DOP=(2000-2023)) AND ALL=("transport" OR "rail" OR "railway" OR "BIM" OR "5D BIM" OR "cost management" OR "cost control" OR "project" OR "cost model" OR "causes" OR "sources" OR "driver" OR "life cycle cost")) AND LA=(English)) NOT DT=(Book OR Book Chapter OR Book Review)

(2—Cost Management and Control)

**link**

https://www.webofscience.com/wos/woscc/summary/67f7e719-745c-487c-9865-b7299555a9f9-694a9ac3/relevance/1 (accessed on 8 January 2023).

Search query string

((((AK=("cost management" OR "cost control" OR "project cost management"OR "cost growth" OR "cost underestimation")) AND PY=(2021-2023)) AND LA=(English)) AND ALL=("transport" OR "rail" OR "railway" OR "BIM" OR "5d BIM" OR "cost model" OR "life

cycle cost" OR "strategies" OR " polices" OR "cost overrun" OR "cost overruns" OR "cost escalation" OR "budget overrun")) NOT DT=(Book OR Book Chapter OR Book Review)

(3—BIM)

**link**

https://www.webofscience.com/wos/woscc/summary/8d726ab0-478b-4fab-81fd-3e006aaf95eb-694a9ef3/relevance/1 (accessed on 8 January 2023).

Search query string

((((AK=("BIM" OR "5d BIM" OR "building information modelling" OR "building information modeling" OR "cost growth" OR "cost underestimation")) AND ALL=("transport" OR "rail" OR "railway" OR "cost management" OR "cost control" OR "cost model" OR "life cycle cost" OR "cost overrun" OR "cost overruns" OR "cost escalation" OR "budget overrun")) AND LA=(English)) AND PY=(2000-2023)) NOT DT=(Book OR Book Chapter OR Book Review)

(4—Rail projects)

**link**

https://www.webofscience.com/wos/woscc/summary/9b8c6cd3-3220-43e6-b8ee-4e82bf7184f8-694aa333/relevance/1 (accessed on 8 January 2023).

**Search query string**

(((AK=("rail" OR "railway" OR "cost growth" OR "cost underestimation")) AND PY=(2000-2023)) NOT DT=(Book OR Book Chapter OR Book Review)) AND ALL=("BIM" OR "5d BIM" OR "cost management" OR "cost control" OR "cost model" OR "life cycle cost" OR "cost overrun" OR "cost overruns" OR "cost escalation" OR "budget overrun")

Science Direct

Science direct only allow max eight boolean connector per field, as a result the search was divided into 2 groups.

* Year 2021–2023

** Exclude book chapters

*** Review articles + Research Article + Short communications

**** English Language

(1—Cost Overrun)

First group:

KEY("cost overrun" OR "cost overruns" OR "cost escalation" OR "budget overrun" OR "cost growth" OR "cost underestimation")

AND

ALL ("transport" OR "rail" OR "railway" OR "BIM" OR "5D BIM" OR "cost management" OR "cost control" OR "project" OR "cost model")

Second group:

KEY("cost overrun" OR "cost overruns" OR "cost escalation" OR "budget overrun" OR "cost growth" OR "cost underestimation")

AND

ALL ("causes" OR "sources" OR "drivers" OR "life cycle cost")

(2—Cost Management and Control)

First group:

KEY ("cost management" OR "cost control" OR "project cost management" OR "cost growth" OR "cost underestimation")

AND

ALL ("transport" OR "rail" OR "railway" OR "BIM" OR "5d BIM" OR "cost model" OR "life cycle cost" OR "strategies" OR " polices"))

Second group:

KEY ("cost management" OR "cost control" OR "project cost management" OR "cost growth" OR "cost underestimation")

ALL ("causes" OR "sources" OR "drivers" OR "life cycle cost")

(3—BIM)

First group:

KEY ("BIM" OR "5d BIM" OR "building information modelling" OR "building information modeling" OR "cost growth" OR "cost underestimation")

AND

ALL ("transport" OR "rail" OR "railway" OR "cost management" OR "cost control" OR "cost model" OR "life cycle cost"))

Second group:

KEY ("BIM" OR "5d BIM" OR "building information modelling" OR "building information modeling" OR "cost growth" OR "cost underestimation")

AND

ALL ("cost overrun" OR "cost overruns" OR "cost escalation" OR "budget overrun")

(4—Rail projects)

First group:

KEY ("rail" OR "railway" OR "cost growth" OR "cost underestimation")

AND

ALL ("BIM" OR "5d BIM" OR "cost management" OR "cost control" OR "cost model" OR "life cycle cost")

Second group:

KEY ("rail" OR "railway" OR "cost growth" OR "cost underestimation")

AND

ALL ("cost overrun" OR "cost overruns" OR "cost escalation" OR "budget overrun")

Google Scholar

The search algorithm for Google scholar is not known and cannot be controlled, Google adapts the search to each user in order to personalize information and, as a result, a systematic search is quite probably not replicable.

Google Scholar was considered as additional source only for this Systematic literature review

* Year 2000–2023

** Search exact phrases in document title only

(1—Cost Overrun)

1. allintitle: Cost overrn model
2. allintitle: Cost overrns model
3. allintitle: cost overruns cause
4. allintitle: cost overruns causes
5. allintitle: cost overrun cause
6. allintitle: cost overruns drivers
7. allintitle: cost overrun drivers
8. allintitle: cost overun transport
9. allintitle: cost overuns transport
10. allintitle: cost overrun rail
11. allintitle: cost overruns rail
12. allintitle: cost overruns life cycle cost
13. allintitle: cost overrun BIM

(2—Cost Management and Control)

1. allintitle: Cost management transport
2. allintitle: cost management rail
3. allintitle: cost management railway
4. allintitle: cost management BIM
5. allintitle: cost management overrun
6. allintitle: cost management overruns
7. allintitle: cost control transport
8. allintitle: cost control rail
9. allintitle: cost control railway
10. allintitle: cost control BIM

11. allintitle: cost control life cycle cost
12. allintitle: project cost management BIM

(3—BIM)

1. allintitle: BIM transport
2. allintitle: BIM rail
3. allintitle: BIM railway
4. allintitle: BIM cost model

(4—Rail projects)

1. allintitle: Rail life cycle cost
2. allintitle: Railway life cycle cost

Search terms and boolean operators

| Scopus | | | | | |
|---|---|---|---|---|---|
| * Year 2000–2023 | | | | | |
| ** English Language | | | | | |
| *** Journal articles, conference papers | | | | | |
| No | Cluster | Keywords | | Research Focus (All fields) | Excluded subject areas |
| 1 | Cost overrun | cost overrun | + | Transport | Earth and Planetary Sciences |
| | | cost overruns | | Rail OR Railway | Mathematics |
| | | cost escalation | | BIM OR 5D BIM | Materials Science |
| | | budget overrun | | Cost Management OR Cost control | Physics and Astronomy |
| | | | | project | Agricultural and Biological Sciences |
| | | | | cost model | Biochemistry, Genetics and Molecular Biology |
| | | | | causes OR sources OR drivers | Psychology |
| | | | - | life cycle cost | Environmental Science |
| | | | | | Medicine |
| | | | | | Chemistry |
| 2 | Cost Management & Control | cost management | + | Transport | Earth and Planetary Sciences |
| | | cost control | | Rail OR Railway | Mathematics |
| | | Project cost management | | BIM OR 5D BIM | Materials Science |
| | | | | cost model | Physics and Astronomy |
| | | | | life cycle cost | Agricultural and Biological Sciences |
| | | | | strategies OR policies | Biochemistry, Genetics and Molecular Biology |
| | | | | cost overrun OR cost overruns OR cost escalation OR budget overrun | Psychology |
| | | | - | | Environmental Science |
| | | | | | Medicine |
| | | | | | Pharmacology, Toxicology and Pharmaceutics |
| | | | | | Chemistry |



| No | Cluster | Keywords | +/- | Research Focus (All fields) | | Subject areas |
|---|---|---|---|---|---|---|
| | | | | | | Health Professions |
| | | | | | | Immunology and Microbiology |
| | | | | | | Neuroscience |
| | | | | | | Nursing |
| | | | | | | Dentistry |
| | | | | | | Veterinary |
| 3 | BIM | BIM | + | Transport | | Earth and Planetary Sciences |
| | | 5D BIM | | Rail OR Railway | | Mathematics |
| | | Building information modelling | | Cost Management OR Cost control | | Materials Science |
| | | | | | | Physics and Astronomy |
| | | | | cost model | | Agricultural and Biological Sciences |
| | | | | life cycle cost | | Biochemistry, Genetics and Molecular Biology |
| | | | | cost overrun OR cost overruns OR cost escalation OR budget overrun | | Environmental Science |
| | | | | | | Medicine |
| | | | | | | Pharmacology, Toxicology and Pharmaceutics |
| | | | | | | Chemistry |
| | | | | | | Immunology and Microbiology |
| | | | | | | Neuroscience |
| 4 | Rail projects | Rail | + | BIM OR 5D BIM | | Earth and Planetary Sciences |
| | | Railway | | Cost Management OR Cost control | | Mathematics |
| | | | | cost model | | Materials Science |
| | | | | life cycle cost | | Physics and Astronomy |
| | | | | cost overrun OR cost overruns OR cost escalation OR budget overrun | | Agricultural and Biological Sciences |
| | | | | | | Biochemistry, Genetics and Molecular Biology |
| | | | | | | Environmental Science |
| | | | | | | Medicine |
| | | | | | | Health Professions |
| | | | | | | Chemistry |
| | | | | | | Neuroscience |
| Total | | | | | | |

| Web of Science | | | | | | |
|---|---|---|---|---|---|---|
| * Year 2000–2023 | | | | | | |
| ** English Language | | | | | | |
| No | Cluster | Keywords | | Research Focus (All fields) | | Excluded subject areas |
| 1 | Cost overrun | cost overrun | + | Transport | - | Environmental Sciences Ecology |
| | | cost overruns | | Rail OR Railway | | Materials Science |
| | | cost escalation | | BIM OR 5D BIM | | Chemistry |

| | | | | | | |
|---|---|---|---|---|---|---|
| | | budget overrun | | Cost Management OR Cost control | | Geography |
| | | | | project | | Physics |
| | | | | cost model | | Mathematics |
| | | | | causes OR sources OR drivers | | |
| | | | | life cycle cost | | |
| 2 | Cost Management & Control | cost management | + | Transport | - | Environmental Sciences Ecology |
| | | cost control | | Rail OR Railway | | Agriculture |
| | | Project cost management | | BIM OR 5D BIM | | Materials Science |
| | | | | cost model | | Health Care Sciences Services |
| | | | | life cycle cost | | Physical Geography |
| | | | | strategies OR policies | | Biomedical social sciences |
| | | | | cost overrun OR cost overruns OR cost escalation OR budget overrun | | Physics |
| | | | | | | Mathematics |
| | | | | | | Biotechnology Applied Microbiology |
| | | | | | | Chemistry |
| | | | | | | Energy fuels |
| | | | | | | Food Science technology |
| | | | | | | Forestry |
| | | | | | | Geology |
| | | | | | | Infectious diseases |
| | | | | | | Metallurgy Metallurgical Engineering |
| | | | | | | Nursing |
| | | | | | | Obstetrics Gynecology |
| | | | | | | Otorhinolaryngology |
| | | | | | | Pharmacology Pharmacy |
| | | | | | | Instrument Instrumentation |
| | | | | | | General Internal Medicine |
| | | | | | | Mechanics |
| 3 | BIM | BIM | + | Transport | | Environmental Sciences Ecology |
| | | 5D BIM | | Rail OR Railway | | Materials Science |
| | | Building information modelling | | Cost Management OR Cost control | | Chemistry |
| | | | | | | Physics |
| | | | | cost model | | Physical Geography |
| | | | | life cycle cost | | Energy fuels |
| | | | | cost overrun OR cost overruns OR cost escalation OR budget overrun | | Remote Sensing |
| | | | | | | Instrument Instrumentation |
| | | | | | | Biotechnology Applied Microbiology |
| | | | | | | Geology |

| No | Cluster | Keyword | + | Research Focus | Subject areas |
|----|---------|---------|---|----------------|---------------|
| | | | | | Imaging Science Photographic Technology |
| | | | | | Robotics |
| | | | | | Agriculture |
| | | | | | Biophysics |
| | | | | | Cell Biology |
| | | | | | Mechanics |
| | | | | | Oncology |
| | | | | | Acoustics |
| | | | | | Astronomy Astrophysics |
| | | | | | Biomedical social science |
| | | | | | Endocrinology Metabolism |
| | | | | | Food science technology |
| | | | | | Geography |
| | | | | | Hematology |
| | | | | | Mechanics |
| | | | | | Neuroscience Neurology |
| | | | | | Optics |
| | | | | | Physiology |
| | | | | | Sociology |
| 4 | Rail projects | Rail | + | BIM OR 5D BIM | Environmental Sciences |
| | | Railway | | Cost Management OR Cost control | Environmental Sturdies |
| | | | | cost model | Material Sciences Multidisciplinary |
| | | | | life cycle cost | Automation control systems |
| | | | | cost overrun OR cost overruns OR cost escalation OR budget overrun | Chemistry Multidisciplinary |
| | | | | | Geography Physical |
| | | | | | Instrument Instrumentation |
| | | | | | Robotics |
| | | | | | Geography |
| | | | | | Energy fuels |
| | | | | | Geoscience Multidisciplinary |
| | | | | | Health Care Sciences Services |
| | | | | | Medical informatics |
| | | | | | Remote sensing |
| Total | | | | | |

| Science Direct | | | | | |
|----|---------|---------|---|----------------|---------------|
| * Year 2000–2023 | | | | | |
| ** Exclude book chapters | | | | | |
| *** Review articles + Research Article + Short communications | | | | | |
| No | Cluster | Keywords | | Research Focus (All fields) | Excluded subject areas |
| 1 | Cost Overrun | cost overrun | + | Transport | - | Medicine and Dentistry |
| | | cost overruns | | Rail OR Railway | | Environmental Science |
| | | cost escalation | | BIM OR 5D BIM | | |
| | | budget overrun | | Cost Management OR Cost control | | |
| | | | | project | | |
| | | | | cost model | | |
| | | | | causes OR sources OR drivers | | |
| | | | | life cycle cost | | |

| 2 | Cost Management & Control | cost management | + | Transport | - | Medicine and Dentistry |
| | | cost control | | Rail OR Railway | | Environmental Science |
| | | Project cost management | | BIM OR 5D BIM | | Agricultural and Biological Sciences |
| | | | | cost model | | |
| | | | | life cycle cost | | |
| | | | | strategies OR policies | | |
| | | | | cost overrun OR cost overruns OR cost escalation OR budget overrun | | |
| 3 | BIM | BIM | + | Transport | - | Medicine and Dentistry |
| | | 5D BIM | | Rail OR Railway | | Environmental Science |
| | | Building information modelling | | Cost Management OR Cost control | | Agricultural and Biological Sciences |
| | | | | cost model | | |
| | | | | life cycle cost | | |
| | | | | cost overrun OR cost overruns OR cost escalation OR budget overrun | | |
| 4 | Rail Projects | Rail | + | BIM OR 5D BIM | | Mathematics |
| | | Railway | | Cost Management OR Cost control | | Environmental Science |
| | | | | cost model | | Psychology |
| | | | | life cycle cost | | |
| | | | | cost overrun OR cost overruns OR cost escalation OR budget overrun | | |
| Total | | | | | | |

| Google Scholar | | | |
|---|---|---|---|
| * Year 2000–2023 | | | |
| ** Document title only | | | |
| No | Cluster | search words (in title only) | |
| 1 | Cost overrun | Cost overrun model | |
| | | Cost overruns model | |
| | | cost overruns cause | |
| | | cost overruns causes | |
| | | cost overrun cause | |
| | | cost overruns drivers | |
| | | cost overrun drivers | |
| | | cost overrun transport | |
| | | cost overruns transport | |
| | | cost overrun rail | |

| | | cost overruns rail | |
|---|---|---|---|
| | | cost overruns life cycle cost | |
| | | cost overrun BIM | |
| 2 | Cost Management & Control | Cost management transport | |
| | | cost management rail | |
| | | cost management railway | |
| | | cost management BIM | |
| | | cost management overrun | |
| | | cost management overruns | |
| | | cost control transport | |
| | | cost control rail | |
| | | cost control railway | |
| | | cost control BIM | |
| | | cost control life cycle cost | |
| | | project cost management BIM | |
| 3 | BIM | BIM transport | |
| | | BIM rail | |
| | | BIM railway | |
| | | BIM cost model | |
| 4 | Rail projects | Rail life cycle cost | |
| | | Railway life cycle cost | |
| Total | | | |

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
