# Peer review of "Minimizing Cost Overrun in Rail Projects through 5D-BIM: A Systematic Literature Review"

_infrastructures, doi:10.3390/infrastructures8050093_

Round 1
Reviewer 1 Report
This extensive review involves the minimizing cost overrun in rail projects though 5D-BIM. This subject is novel, interesting and new. The literature references (343 articles) are adequate. In general, this review is well-written. Please, the following changes/improvements are suggested.
- A careful check of grammar and syntax is necesssary, because the overall language is good, but some points should be checked again in the whole text. Also, please check the gaps between words, or synthetic words. Replace the "&" with "and " in the whole text.
- In line 32, where did you find the 44.7 % ?
- In the introduction, please give more emphasis on rail projects.
- In Table 1, please correct, the numbering of lines is in the first column, and the title of the table shall be together with the table.
- In Figs. 2,13 why do you repeat this from the literature? do you have the right? Have you checked plagiarism ?
- In Figs. 3,15,16, the letters are too small.
- Please, can you add some details, on how you constructed Fig. 8 ?
- What is the meaning of Fig. 9 ? this figure is so blurred, that no meaning is possible.
- The table 4, figure 12 should have portrait layout.
- In line 424, correct the format, to understand that this is a new Section. Similarly, in line 693,
- The Sections 4.5 and 4.6 are a little complicated, with a a medium syntax and format of sentences, and present lower quality than the document's first part. Please, improve your writting at the medium and last part of your document.
- All tables should have their title together with the table.
- The format of title text in all figures should follow the journal template.
- Better avoid bold letters in lines 715-743.
- The conclusions should be reviewed in more mature way.
- There are a lot of abbreviations within the whole text, perhaps a table with abbreviations should be added.
- The format of references, at the end of this documents should be corrected to the journal template.
- In the appendices, all tables are oriented in landscape, which is in contrast to the rest of the document. Please, is it possible to change the tables in portrait layout?
The English language needs revision, as mentioned above.
Reviewer 2 Report
This manuscript contains a systematic literature review dedicated to rail projects and 5D-BIM. The methodology used in this review is impressive and very well characterized. Detailed descriptions and the addition of the search procedures will be helpful to other researchers with other reviews independent from the topics. So, I evaluate high this aspect of the study.
The use of such a methodology for analyzing the title topic (Minimizing Cost Overrun in Rail Projects Through 5D-BIM) is also effective. So, the research questions formulated in the introduction (lines 49 – 58) are solved satisfactorily. The answers to these questions are clearly presented in the conclusions (lines 760 – 796). These conclusions will be helpful in future rail projects with the potential use of the 5D-BIM methodology. This aspect of the study is also a strong part of the manuscript.
I found small technical (graphical) problems only. And I request for corrections:
- Figures 6 and 7, “no of publications…” suggests the lack of them, the better way is to use the full word “number”.
- The content of figures 9, 10, 11, 15, and 16 are weakly readable, probably the use of bigger and black fonts will delete this problem.
- The sentence “Figures Figure 15 and Figure 16” (line 715) should be corrected (deleting multiplications of the word "figure").
After these corrections, I recommend publishing this interesting paper.
Minor editing of English language required
Reviewer 3 Report
The manuscript entitled “Minimizing Cost Overrun in Rail Projects Through 5D-BIM: A Systematic Literature Review” proposes a systematic literature review on the state-of-the-art implementation of 5D-BIM for cost control and management in rail infrastructure projects. The paper is interesting and well-written, however, these are some topics that I believe could be improved.
MAJOR COMMENTS:
1) INTRODUCTION: In my opinion, the Introduction section should provide a fast initial understanding of the research. For this purpose, it must consist of a brief introduction to the topic, delimit the research gap, the objective, highlight the novelty of the research, and provide a brief explanation of the methodology to be used. Even review papers should follow this pattern, so that the reader can be informed about the main aspects of the article right at the beginning of the reading. The current text somewhat touches on some of these aspects, but in my opinion, it can be improved. Mainly regarding the SLR methodology to be used, the delimitation of the research gap, and the novelty (Why is this review different from the ones that have already been conducted? Is it the first?)
2) SECTION 3.1.3: Authors must provide the Boolean operators used in the searches in each of the databases so that the study can be replicated. Besides defining the publication period of the studies, was any other type of filter used during the search? For example, source type, language, knowledge area, etc...
3) SECTION 3.1.3. Lines 349-352: I suggest that authors indicate why they chose these five tags. Have they been used by other authors? Have the authors defined based on their own experience? There is no right or wrong answer, it's just a matter of transparency about the adopted parameters.
4) SECTION 3.1.3. Lines 338-357: The explanation provided by the text of these four paragraphs does not follow what is shown in the diagram in Figure 4. After the duplicate removal step, the authors do not explain the parameters adopted for excluding the 1,358 records after reviewing the titles and abstracts. In other words, the explanation goes directly to the information that 1,888 documents were considered in the eligibility first step. Please indicate why these 1,358 documents were excluded.
5) SECTION 3.1.3. In the second round of the eligibility step, what criteria were used to consider 1,524 documents as not relevant for the SLR? What changed from the first to the second round?
6) FIGURE 5: The figure must be fully edited. It is not possible to fully read any of the texts within the Figure. It is also not possible to understand the subdivision of each tag based on document types (journal, conference, etc.), as it is not possible to identify which part of the diagram is related to each one. Authors should think in a way to present the texts more adequately, or else reformulate the system of colors and legends.
7) FIGURE 5: I don't understand why this figure is positioned in Section 3. It presents the results of the SLR process and as such should be placed in the Results Section.
8) SECTIONS 4.1 to 4.4: In my opinion, the division of the text into several subsections composed of only one small paragraph is not interesting. I suggest writing a more unified text. In addition, as this is a results and discussion section, I suggest that the data presented in the graphs are better discussed in the text. For example, Figure 6 shows a jump in publications in a given year, as well as a trend toward an increase in publications on the topic over the past few years. What could be the reasons for this? In the case of Figure 7, it can be seen that two journals have quite relevant publication numbers. Why? Do they feature special issues on the topic? Is the topic part of the permanent scope of the journal?
9) FIGURE 6: The figure title indicates that articles published in 2023 were not counted. In that sense, I suggest updating the information presented in line 310: “...on the period between 2000 and 2023”. In fact, the study considered the period from 2000 to 2022.
10) FIGURES 6, 7, and 10: I don't understand why these figures consider the 1,888 documents since only 380 passed the selection process. Typically, graphs and statistics are created based on selected articles. Authors must clearly present at what stages the documents were considered.
11) A sentence is not considered a paragraph. Check all the text and make sure the sentences can be condensed to form real paragraphs.
MINOR COMMENTS:
12) Line 77 and Line 83: Use only one currency type. This allows the reader not to be confused by the value entered after the conversion. Please check the rest of the text.
13) FIGURE 2: In my opinion, this figure is unnecessary, as it does not add new information beyond that already provided in the text.
14) SECTION 3: I suggest changing the heading of this section to “Materials and Methods” or something similar.
15) FIGURES 5, 6, 7, 9, 10, 13, 15, and 16: I suggest editing these figures. It is not necessary to include titles in the graphics print area, as they are already indicated in the title of the figures. The presence of gray borders also pollutes the text.
16) FIGURE 6: It may be interesting, within the bar graph itself, to divide the different types of publications (journal, conference, etc.) by year.
17) FIGURE 7: The name of the journals should be presented in full in the figure.
18) FIGURE 8: I'm not sure the figure title is interesting. The word "frequency" conveys the idea of volume of publications over time, however, from what I understand, the figure shows the total number of publications per country in the analyzed period. Please check! I also suggest increasing the figure scale since there is available space on the page for this.
19) FIGURE 9: I understand that the program used has limited options for network visualization. However, the authors need to work on the Figure, as the letters are so small that it is virtually impossible to visualize the results. Perhaps the definition that only two publications were enough to constitute a network favored the formation of networks of little relevance (few authors or few publications), generating distortions.
Round 2
Reviewer 1 Report
The authors have improved this paper, according to the comments of the reviewers. This paper is complete.
(Please note that some tables remain with landscape orientation in paper. )
This paper can be published. Please check if the landscape orientation in some tables in the manuscript is allowed by the journal template.
Reviewer 3 Report
---